# MRConv: Reparameterized Multi-Resolution Convolutions for Long Sequence Modelling

**Harry Jake Cunningham**[*]
University College London

**Giorgio Giannone**[†]
University College London
Amazon

**Mingtian Zhang**
University College London

**Marc Peter Deisenroth**
University College London

## Abstract

Global convolutions have shown increasing promise as powerful general-purpose sequence models. However, training long convolutions is challenging, and kernel parameterizations must be able to learn long-range dependencies without overfitting. This work introduces reparameterized multi-resolution convolutions (`MRConv`), a novel approach to parameterizing global convolutional kernels for long-sequence modelling. By leveraging multi-resolution convolutions, incorporating structural reparameterization and introducing learnable kernel decay, `MRConv` learns expressive long-range kernels that perform well across various data modalities. Our experiments demonstrate state-of-the-art performance on the Long Range Arena, Sequential CIFAR, and Speech Commands tasks among convolution models and linear-time transformers. Moreover, we report improved performance on ImageNet classification by replacing 2D convolutions with 1D `MRConv` layers.

## 1 Introduction

Modelling sequences with long-range dependencies is critical for solving tasks such as time-series forecasting, speech recognition and language modelling. Numerous deep learning models have been designed to address this challenge by aggregating information across long contexts, including recurrent neural networks (RNNs) [2, 28], convolutional neural networks (CNNs) [37, 4, 5, 40], and Transformers [46]. Despite significant progress, these models still struggle to effectively model long sequences due to training instabilities, inadequate inductive biases to prioritize important contextual information and prohibitive computational complexities.

Recently, State Space Models (SSMs) [22, 23, 43, 25, 19] have demonstrated their ability to model extremely long sequences. SSMs corresponding to a linear time-invariant (LTI) system, can be efficiently implemented using a global depthwise convolution. The convolution kernel is parameterized according to the HiPPO framework by initializing specially constructed state matrices [20, 24]. However, despite their success, SSMs are complex models that rely on sophisticated mathematics and linear algebra to compute the convolution kernel, which for S4 [22] and S4D [23] becomes a bottleneck for the faster downstream convolution.

Inspired by their success and equivalence to global convolutions, several authors have aimed to reduce the complexity of SSMs by *parameterizing long convolution kernels directly* [40, 27, 17, 38, 29]. In general, explicitly parameterized long convolution kernels are extremely difficult to train and are

---

[*]Correspondence to `jake.cunningham.21@ucl.ac.uk`
[†]Work done at University College London.

prone to overfitting, resulting in a significant performance drop compared to SSMs. To ease training, solutions to the parameterization problem have largely focused on: 1) low-rank approximations to the convolutional kernel through implicit neural representations [40, 27, 38], the composition of multi-resolution sub-kernels [29, 42], or regularization [17] and 2) a decaying kernel structure such that weights closer to the input are larger than ones further away [29].

In this work, we introduce reparameterized multi-resolution convolutions (MRConv), a method for parameterizing global convolutional kernels for long-sequence modelling, that simplifies, outperforms, and is more efficient than SSMs. Building on previous work, we focus our attention on **structured multi-resolution sub-kernels**, **low-rank kernel parameterizations** and **learnable kernel decay**. We adopt the multi-resolution approach introduced by SGConv [29], constructing large kernels as the sum of smaller sub-kernels of increasing length but fixed numbers of parameters. To improve performance we use ideas from computer vision, namely *structural reparameterization*. Specifically, to diversify optimization we train each sub-kernel in parallel, summing their activations after batch normalization, before merging all parameters into a single convolution at inference. We also explore several different low-rank kernel parameterizations and how these can be linearly combined during training to learn expressive long-range kernels that perform well across a wide range of data modalities.

Our contributions are summarized as follows:

- Inspired by SGConv [29], our MRConv structure constructs global kernels as the learnable combination of low-rank sub-kernels of increasing length but equal numbers of parameters, each designed to model the input at a different resolution and protect from overfitting.

- MRConv uses a novel reparameterization scheme, allowing each sub-kernel to be trained in parallel, utilizing batch normalization and linear scaling to learn the kernel decay rate, before merging them into a single global convolution at inference.

- We demonstrate that MRConv achieves **state of the art performance** among attention-free models and linear-time Transformers across various data modalities on Long Range Arena, sCIFAR, Speech Commands and ImageNet, whilst also improving efficiency.

## 2 Background

**State Space Models** We consider discretized, linear time-invariant (LTI) state-space models (SSMs) of the form

$$x_t = \boldsymbol{A}x_{t-1} + \boldsymbol{B}u_t \tag{1}$$

$$y_t = \boldsymbol{C}x_t + \boldsymbol{D}u_t, \tag{2}$$

where $x_t \in \mathbb{R}^D$ is the hidden state at time $t = 1, \ldots, L$, $u_t \in \mathbb{R}$ is a scalar input signal, and $y \in \mathbb{R}$ is an output signal. Essential to the success of SSMs is initialization of the system matrices $\boldsymbol{A} \in \mathbb{R}^{D \times D}$, $\boldsymbol{B} \in \mathbb{R}^{D \times 1}$, $\boldsymbol{C} \in \mathbb{R}^{1 \times D}$, and $\boldsymbol{D} \in \mathbb{R}^{1 \times 1}$ according to the HiPPO theory that projects the input sequence onto a set of orthogonal polynomials equipped with exponential decay [20, 24].

Unrolling the recursion over the length of the time horizon $L$, the output $y \in \mathbb{R}^L$ can equivalently be computed as a 1D causal convolution, avoiding computation of the hidden states, which can become very memory intensive, as

$$\boldsymbol{K} = \begin{bmatrix} \boldsymbol{C}\boldsymbol{B}, \boldsymbol{C}\boldsymbol{A}\boldsymbol{B}, \cdots, \boldsymbol{C}\boldsymbol{A}^{L-1}\boldsymbol{B} \end{bmatrix} \tag{3}$$

$$y = \boldsymbol{K} * u + \boldsymbol{D}u, \tag{4}$$

where $u = [u_1, \ldots, u_L]^T \in \mathbb{R}^L$. Computing the convolutional kernel for S4 and S4D scales as $\mathcal{O}((L+D)\log^2(L+D))$ using fast Cauchy and Vandermonde matrix-vector products, although this bottlenecks the faster $\mathcal{O}(L\log L)$ convolution implemented using Fast Fourier Transforms (FFTs).

**Convolutional Models for Sequence Modelling** Interpreting SSMs as a global convolution implicitly parameterized by the system matrices, we can also consider alternative implicit parameterizations. In general, implicit parameterizations define kernel values $\boldsymbol{K}[t]$ as a function of the filter locations $t = 1, \ldots, L$,

$$\boldsymbol{K}[t] = \alpha^{-t} f_\theta(t), \tag{5}$$

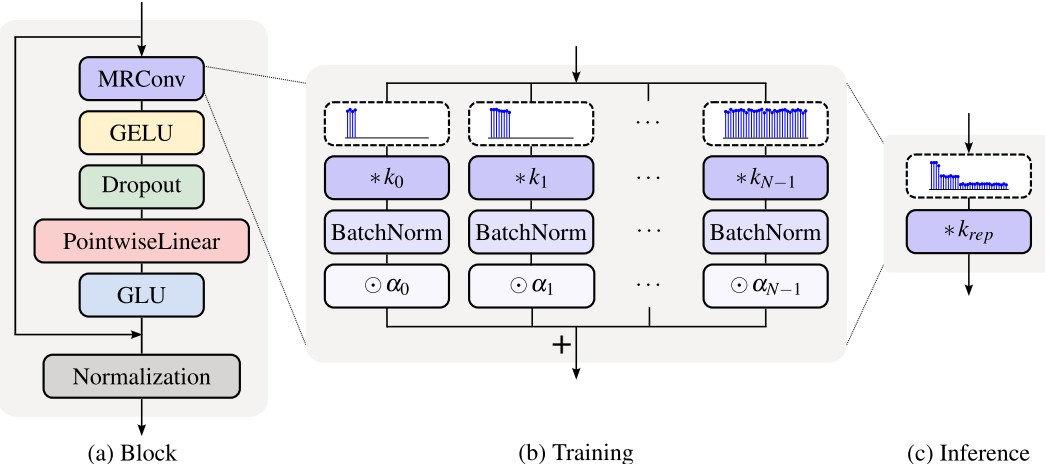

(a) Block                    (b) Training                    (c) Inference

Figure 1: **Left**: The `MRConv` block is composed of a `MRConv` layer, GELU activation, pointwise linear layer, to mix the channels, and a gated linear unit. **Middle**: During training, the `MRConv` layer processes the input using $N$ *branches* each with it's own convolution kernel of increasing length and BatchNorm parameters. The output of the layer is given by pointwise multiplying each branch by $\alpha_i$ and summing. **Right**: At inference the branches can be reparameterised into a single convolution.

where $f_\theta$ is a parametric function with parameters $\theta \ll L$ and $\alpha$ is some decay constant. Several parameterizations of $f_\theta$ have been proposed including MLPs [40, 27, 38] and linear interpolation [29]. The low-rank structure of implicit parameterizations, coupled with exponential decay has proven a successful inductive bias for long-sequence modelling, ensuring the magnitude of kernel weights is greater for near information and improving generalization by preventing overfitting on irrelevant long-range dependencies.

## 3 Reparameterized Multi-Resolution Convolutions

We propose `MRConv`, a set of depthwise separable multi-resolution convolutions designed for long-sequence modelling. Addressing the need for implicit kernel parameterizations with a decay structure, we construct global kernels as the *learnable summation* of normalized multi-resolution sub-kernels of increasing length but with a constant number of parameters, using more parameters to aggregate local information. Further, we use *causal structural reparameterization* to train individual sub-kernels in parallel for diverse optimization and improved model performance, before combining them into a single kernel for efficient inference. See Figure 1 for an overview of the `MRConv` block.

In Section 3.1, we define 1D causal structural reparameterization and in Section 3.2 we introduce our reparameterization scheme for merging multi-resolution convolutions. In Section 3.3, we introduce 3 kernel parameterizations for generating low-rank kernels with fixed numbers of parameters but variable lengths.

### 3.1 Causal Structural Reparameterization

Our multi-resolution strucutre is based on the linearity of the convolution operator, which allows us to merge multiple branches of causal convolutions into a single convolution as

$$y[t] = \sum_{n=0}^{N-1} (u * k_n)[t] = \left( u * \left( \sum_{n=0}^{N-1} k_n \right) \right) [t] = (u * k_{rep})[t], \tag{6}$$

where $k_n$ is the convolution kernel of the $n$th branch and $k_{rep} = \sum_{n=0}^{N-1} k_n$ is the *reparameterized kernel* computed by summing all $n$ kernels together. In order to reparameterize causal convolution kernels of different sizes, we must ensure that the kernels are correctly aligned spatially before summing them, such that $k_n[\tau]$ for all kernels acts on the input at $u[t - \tau]$. To do this we pad shorter kernels of length $l$ to the right with zeros such that the length of the kernel is the length of the longest

kernel $L$ and then sum, such that

$$k_{rep} = \sum_{n=0}^{N-1} \text{ZeroPad}(k_n, (0, L-l)) = \sum_{n=0}^{N-1} \bar{k}_n, \tag{7}$$

where $\bar{k}_n$ corresponds to the zero-padded version of $k_n$. In this work, we consider two types of structural reparameterization: 1) *causal branch addition with batchnorms* for combining causal convolutions of varying length, and 2) *causal branch addition with linear rescaling* for combining causal convolutions of equal length.

**Causal Branch Addition with BatchNorm**  When merging kernels of different lengths, normalization of each branch becomes crucial due to the impact of kernel size on the output statistics of convolutions with different length kernels. Hence, when constructing multi-resolution branches we use BatchNorm after each convolution to normalize the features, which can then be merged into the preceding convolution layer at inference via

$$k_{rep} = \overline{\text{BN}_0(k_0)} + \overline{\text{BN}_1(k_1)}, \tag{8}$$

where the lengths of the normalized kernels are adjusted according to equation 7.

**Causal Branch Addition with Linear Rescaling**  When merging kernels of the same length normalization is unnecessary. In [26], the authors argue that the scaling factors of norm layers matter most as they diversify the optimization of different branches and instead propose replacing non-linear norm layers with linear scaling layers that can be reparameterized during training. We follow this advice for merging kernels of equal length as

$$k_{rep} = \beta_0 \cdot k_0 + \beta_1 \cdot k_1. \tag{9}$$

This reduces both memory and computational costs during training as all layers are now linear and hence kernels can be reparameterized during training.

## 3.2  Multi-Resolution Convolutions

We now outline our reparameterization scheme as a means of training multi-resolution convolutions using branches that can be combined into a single convolution kernel for efficient inference. Let $u \in \mathbb{R}^{D \times L}$ be a $D$-dimensional input sequence of length $L$. We define the number of independent resolutions as $N = \log_2(L/l_0) + 1$ where $l_0$ is the size of the kernel at the first resolution. At each resolution $i$, we define a kernel $k_i$ of length $l_i = l_0 2^i$ for all $i < N$. We denote the output of each convolution as $c_i = (k_i * u) \in \mathbb{R}^{D \times L}$. Following each convolution, we pass the output through a BatchNorm, $\tilde{c}_i = \text{BN}_i(c_i)$, where each resolution has its own set of BN parameters. We define the set of normalized multi-resolution convolution outputs $\tilde{c} \in \mathbb{R}^{N \times D \times L}$ as,

$$\tilde{c} = [\text{BN}_0(k_0 * u), \text{BN}_1(k_1 * u), \cdots, \text{BN}_{N-1}(k_{N-1} * u)]. \tag{10}$$

Given $\tilde{c}$ we wish to combine the outputs to generate an output $y$, ensuring the most relevant parts of the input are highlighted. The output $y[t] \in \mathbb{R}^D$ at time step $t$ is generated by computing a linear combination of the coefficients $\tilde{c}[t]$ at time step $t$ according to

$$y[t] = \boldsymbol{\alpha}^T \tilde{c}[t], \tag{11}$$

where $\boldsymbol{\alpha} \in \mathbb{R}^{N \times D}$ is a learnable parameter. Applying $\boldsymbol{\alpha}$ across the sequence length we define the output $y \in \mathbb{R}^{D \times L}$ as the summation

$$y = \alpha_0 \text{BN}_0(k_0 * u) + \alpha_1 \text{BN}_1(k_1 * u) + \cdots + \alpha_{N-1} \text{BN}_{N-1}(k_{N-1} * u). \tag{12}$$

Further, applying our reparameterization scheme at inference, we can rewrite the above process as a single convolution by zero-padding shorter kernels and merging the BN parameters into each convolution as

$$y = u * (\alpha_0 \overline{\text{BN}_0(k_0)} + \alpha_1 \overline{\text{BN}_1(k_1)} + \cdots + \alpha_{N-1} \overline{\text{BN}_0(k_{N-1})}) = u * k_{rep}, \tag{13}$$

eliminating the extra memory and computational cost of training with extra convolutions.

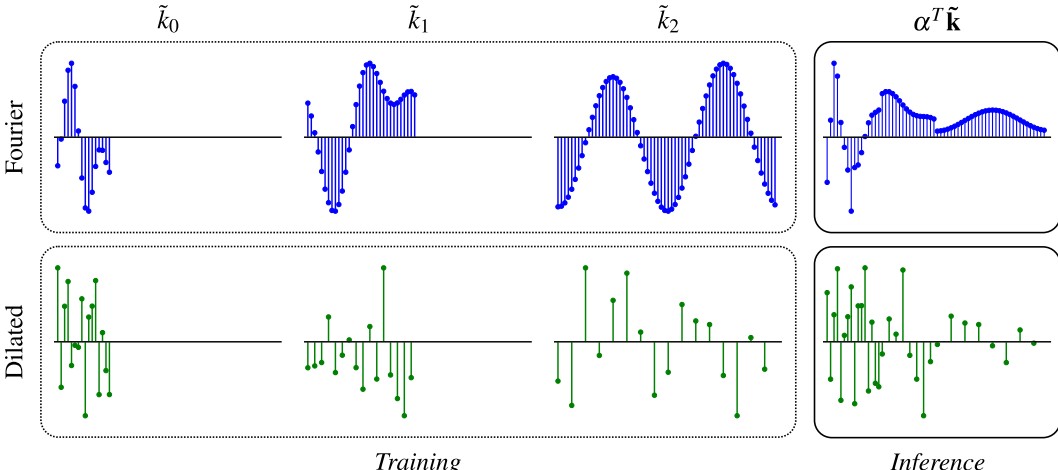

Figure 2: **Multi-resolution structural reparameterization.** During *training*, we parameterize each branch with a kernel of increasing length but fixed number of parameters. For the Fourier kernels, we use only a handful of low-frequency modes and for the dilated kernels we increase the dilation factor. At *inference*, we combine the kernels into a single kernel by merging the BN parameters with the kernel parameters and performing a learnt weighted summation.

### 3.3  Low-Rank Kernel Parameterization

Our multi-resolution framework is general and agnostic to the parameterization of the kernels at each resolution. In this work, we consider 3 different parameterizations:

**Dilated Kernels**  Inspired by wavelets, dilated convolutions are a variation on standard convolutional filters where $p$ many zeros are padded between the elements of the kernel, where $p + 1$ is known as the dilation factor. Formally, dilated convolutions are defined as

$$y[t] = (u * k_{dilated})[t] = \sum_{\tau=0}^{l-1} k[\tau]u[t - p\tau]. \tag{14}$$

They are a parameter-efficient way of increasing the receptive field of a convolution by detaching the length of the kernel from the number of parameters.

**Fourier Kernels**  Instead of parameterizing kernels in the time domain $k \in \mathbb{R}^{D \times L}$, we instead parameterize them as complex valued kernels $\hat{k} \in \mathbb{C}^{D \times L}$ in the Fourier domain. To get $k$ we simply take an inverse Fourier transform of $\hat{k}$, $k = \mathcal{F}^{-1}[\hat{k}]$. We can also generate long low-rank kernels by only parameterizing a small number $m$ of low-frequency Fourier modes. In practice we use FFTs and zero-padding to achieve this, at a cost of $\mathcal{O}(L \log L)$, which is cheaper than computing the kernel in the SSM formulation,

$$k_{fourier}[t] = \text{IFFT}[\text{ZeroPad}(\hat{k}, L - m)])[t]. \tag{15}$$

**Sparse Kernels**  Similar to dilated kernels we also propose sparse kernels, where, instead of regularly spacing kernel elements at a set distance apart, we randomly sample their positions across the sequence length, which we then fix during training and inference. Given a set of kernel value locations $\mathcal{T}$ we define sparse kernels as

$$k_{sparse}[t] = \delta_{t \in \mathcal{T}} \cdot k_t, \tag{16}$$

where $\delta_t \in \mathcal{T}$ is the Kronecker delta, which equals 1 if $t$ is in the set $\mathcal{T}$ and 0 otherwise, and $k_t$ represents the non-zero kernel value at position $t$.

### 3.4  FFT Convolutions

Other than dilated kernel convolutions, which can be computed in $\mathcal{O}(kL)$ using the implicit GEMM algorithm [6], for all kernels we compute the depthwise convolution using FFTs reducing the

computation to $\mathcal{O}(L \log L)$ time complexity. We also make use of a number of highly optimized FFT convolution implementations, which further speeds up our work and reduces memory [17, 18].

**Remark.** *The time complexity of a multi-resolution convolution on a sequence of length $L$ is at most $\mathcal{O}((\log(L/k_0)+1)L \log L)$ during training and $\mathcal{O}(L \log L)$ during inference where each convolution is performed in the Fourier domain.*

## 4 Related Work

Several prior works have used multi-resolution convolutions for general sequence modelling. SGConv [29] is most similar to our approach, using a weaker form of reparameterization, by concatenating smaller low-rank sub-kernels to construct long convolution kernels. Each sub-kernel is implicitly defined by linearly interpolating dilated kernel values and a fixed kernel decay is used. We expand considerably upon their work, exploring several new reparameterization schemes, introduce improved kernel parameterizations and add learnable kernel decay.

MultiresNet [42] uses dilated convolutions with shared weights to parameterize a learnable wavelet transform and also learns a linear combination of outputs similar to our method. However, they don't consider reparameterizing their model into a single global convolution for efficient inference. Ding et al. [14] design a set of structurally parameterized kernels also using dilated kernels and parallel branches during training, however they only consider small kernel sizes and doesn't consider any kernel decay.

Recently CHELA [33] proposed to use short and long convolutions placed sequentially in conjunction with self-attention. However, in their work, in instances where they place a non-linear activation function between the convolutions, the non-linearity restricts the ability to reparameterize both convolutions into a single kernel.

## 5 Experiments

We now evaluate the empirical performance of `MRConv` against similar baseline methods on long sequence modelling tasks. We test 3 different kernel parameterizations: 1) dilated kernels, 2) Fourier kernels and 3) Fourier + sparse kernels, which we reparameterize during training using linear rescaling (see Equation 9). We select model hyperparameters to ensure similar computational complexities to comparable models such as S4 and SGConv. Our results show that `MRConv` is a highly effective and efficient sequence modeller, achieving SoTA performance on LRA, sCIFAR and Speech Commands, whilst being more efficient than SSMs, such as S4, and linear-time transformers.

### 5.1 Long Range Arena

The Long Range Arena (LRA) benchmark [44] evaluates the performance of sequence models on long-range modelling tasks on a wide range of data modalities and sequence lengths from 1,024 to 16,000. For all LRA tasks, we use the standard S4 block (see Figure 1) and use `MRConv` as a drop-in replacement for the SSM layer. We train two model variants: 1) *Base* has similar complexity and parameters to existing convolutional and SSM baselines and 2) *Large* where we scale the model with increased width or depth to match the computational budget set by more expensive quadratic attention baselines (see Table 4b).

Table 1 compares `MRConv` to other baseline methods. Treating the kernel parameterization as a model hyperparameter and selecting the model with the highest validation accuracy, `MRConv`-*Base* achieves the highest average score among sub-quadratic complexity models including S5, SGConv and modern linear-time transformer architectures such as MEGA-Chunk. Further, `MRConv`-*Large* matches the performance of more computationally expensive quadratic transformers whilst being much faster at inference due to reparameterization (see Table 4b). We conduct a series of ablation studies to assess the effectiveness of our kernel parameterizations, reparameterization scheme, and the importance of learnable decay. Additional implementation details can be found in Appendix D.3.

**Kernel Parameterization**   Table 1 displays the LRA results for each proposed kernel parameterization. Dilated kernels perform exceptionally well on the Image task, outperforming all other non-input-dependent models by 1.2%. However, on information-dense tasks such as ListOps, Fourier

Table 1: **Test accuracy on the Long Range Arena Benchmarks**. We follow the standard training procedures introduced in [23]. Bold scores indicate the highest performing model on a given task and underlined the second best performing. ✗ indicates a model did not do better than random guessing and - that a result was not available. In this table we only include results from other non-input-dependent models.

| Model (Input length) | ListOps (2,048) | Text (4,096) | Retrieval (4,000) | Image (1,024) | Pathfinder (1,024) | Path-X (16,384) | Avg. |
|---|---|---|---|---|---|---|---|
| Transformer | 36.37 | 64.27 | 57.46 | 42.44 | 71.40 | ✗ | 53.66 |
| Transformer + SPT | 59.15 | 88.81 | 90.38 | 76.00 | 88.49 | 88.05 | 81.81 |
| *Linear-Time Transformers:* | | | | | | | |
| BST | 61.49 | 87.63 | 90.51 | **91.07** | 95.75 | 95.28 | 86.96 |
| SPADE-Chunk | 60.50 | **90.69** | 91.17 | 88.22 | 96.23 | 97.60 | 87.40 |
| MEGA-Chunk | 58.76 | 90.19 | 90.97 | 85.80 | 94.41 | 93.81 | 85.66 |
| *State Space Models:* | | | | | | | |
| S4D-LegS | 60.47 | 86.18 | 89.46 | 88.19 | 93.06 | 91.95 | 84.89 |
| S4-LegS | 59.60 | 86.82 | 90.90 | 88.65 | 94.20 | 96.35 | 86.09 |
| Liquid-S4* | **62.75** | 89.02 | 91.20 | 89.50 | 94.8 | 96.66 | 87.32 |
| S5 | 62.15 | 89.31 | 91.40 | 88.00 | 95.33 | 98.58 | 87.46 |
| *Convolutional Models:* | | | | | | | |
| CCNN | 43.60 | 84.08 | - | 88.90 | 91.51 | ✗ | - |
| Long Conv | 62.2 | 89.6 | 91.3 | 87.0 | 93.2 | 96.0 | 86.6 |
| SGConv | 61.45 | 89.20 | 91.11 | 87.97 | 95.46 | 97.83 | 87.17 |
| *Ablations:* | | | | | | | |
| MRConv-*B*, Dilated | 60.90 | 86.38 | 88.30 | 90.37 | 94.42 | ✗ | 78.40 |
| MRConv-*B*, Fourier | 62.40 | 89.26 | 91.44 | 88.55 | 95.03 | 97.82 | 87.42 |
| MRConv-*B*, Fourier+Sparse | 62.10 | 89.26 | 91.35 | 89.07 | 95.55 | ✗ | 79.56 |
| MRConv-*L*, Dilated | 61.25 | 88.36 | 89.78 | 90.55 | 95.22 | ✗ | 79.19 |
| MRConv-*L*, Fourier | 62.45 | 89.40 | **91.48** | 89.30 | 95.75 | **98.65** | 87.84 |
| MRConv-*L*, Fourier+Sparse | 61.65 | 89.42 | 91.35 | 89.15 | **96.64** | ✗ | 79.70 |
| *Ours:* | | | | | | | |
| MRConv-*B* | 62.40 | 89.26 | 91.44 | 90.37 | 95.55 | 97.82 | 87.81 |
| MRConv-*L* | 62.45 | 89.42 | **91.48** | 90.55 | **96.64** | **98.65** | **88.20** |

kernels perform better, as the sparse dilated pattern is prone to skipping important tokens. Fourier kernels perform best on average and are the only model that achieves better than random guessing on Path-X, where sparse and dilated kernels struggle due to their sparsity. The combination of sparse and Fourier kernels is also effective on natural data and performs well on Pathfinder but can lead to overfitting on discrete data modalities, such as ListOps.

**MRConv Design**   Table 2 shows that all of our structural additions improve performance over dense kernels by 13.2% and 6.4% on the ListOps and Image tasks. We found that BatchNorms are crucial for learning the linear combination of each multi-resolution convolution, resulting in an improvement in accuracy of 5.35% and 2.36% on both tasks than without them. Interestingly, parameterizing our multi-resolution architecture with dense kernels reduces overfitting, improving the test accuracy by 3.90% and 3.75%, despite having more parameters. Path-X was the only task where we found learning the combination of multi-resolution convolutions did not improve performance and we provide extra implementation details and ablations in Appendix D.3.3.

**Resolution Analysis**   Figure 3b shows the normalized magnitude of the weight $\alpha_i$ for each kernel $k_i$ at different depths of the model after training on ListOps and CIFAR datasets. The resulting weight distribution shows that early layers learn short high-frequency kernels for local features, while deeper layers learn an even distribution across a range of longer kernels. On CIFAR, deeper layers learn low-frequency global kernels, aligning with existing observations on larger models [48]. These non-stationary filter characteristics with depth emphasise the need for effective learnable kernel decay, which is difficult to achieve with hand-tuned initializations such as in SGConv.

## 5.2   Pixel-Level 1D Image Classification

Next, we evaluate `MRConv` on the sequential CIFAR (sCIFAR) image classification task, where images are flattened to a 1D sequence of pixels. This is a challenging sequence modelling task as the model

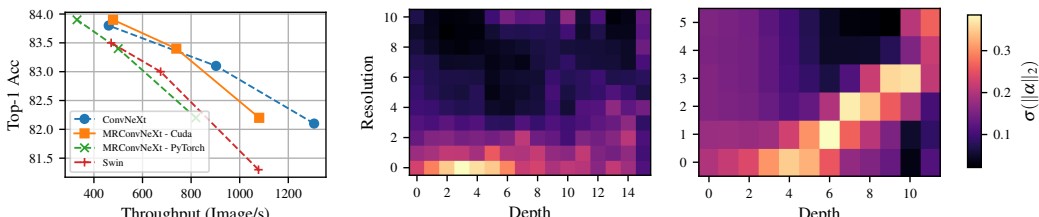

Figure 3: **Left**: ImageNet Top-1 Acc. vs. Throughput. **Right**: Distribution of $\alpha$ norms for each depth for MRConv trained on ListOps and CIFAR respectively. Changing composition of kernels highlights how the convolution kernels are non-stationary with respect to depth.

| Model modification | | ListOps | | | Image | |
| | Params | Accuracy | Change | Params | Accuracy | Change |
|---|---|---|---|---|---|---|
| Dense kernel | 2.4M | 49.25 | - | 6.3M | 82.90 | - |
| + Multiresolution | 4.5M | 53.15 | +3.90 | 9.4M | 86.65 | +3.75 |
| + Fourier kernel | 307K | 57.05 | +7.80 | 3.8M | 86.19 | +3.29 |
| + BatchNorm | 332K | 62.40 | +13.15 | 3.8M | 88.55 | +5.65 |
| + 2x Depth (MRConv, Fourier) | 661K | **62.45** | +13.20 | 7.7M | **89.30** | +6.40 |

Table 2: **MRConv design ablations**. Effect of `MRConv` modifications on ListOps and Image tasks from LRA. For reference we note that *S4-LegS* uses 815K and 3.6M parameters and *Liquid-S4* uses 333K and 11M parmeters for each task respectively.

must be able to capture pixel-level relationships at different scales, as pixels close together in 2D space can be far apart in its flattened sequence representation.

Table 3a reports the performance of `MRConv` compared to other baseline methods. On sCIFAR `MRConv` with dilated kernels significantly improves test accuracy upon the previous best model, MultiresNet, by 1.1%. Further, `MRConv` with dilated kernels is very parameter efficient using 10 layers and 5.7M parameters, in contrast to S4 which uses 6 layers and 7.9M parameters [22]. We also note a significant performance gap between Fourier and dilated parameterizations. We hypothesise that dilated convolutions enhance the ability of long kernels to focus on long-range sparse patterns (i.e. relationships between pixels far apart might be more important than pixels closer together), which is well suited for flattened image data which has a high correlation between neighboring pixels.

### 5.3  Raw Speech Classification

The Speech Commands (SC) dataset [47] contains 1s sound recordings, sampled at 16,000 Hz, of 35 spoken words in English. The task is to classify the spoken word from its sampled waveform. We also test zero-shot classification at a lower sampling rate of 8,000 Hz to test the continuous-time parameterization of each model.

Table 3b reports the results. Both the dilated and Fourier kernel parameterizations perform well, especially `MRConv` equipped with Fourier kernels, which outperforms all baseline models. On the zero-shot task, `MRConv` with Fourier kernels also performs the best. By parameterizing the kernels in the Fourier domain with a set of low-frequency modes, we ensure that the kernels are band-limited. As a result, we can downsample each kernel whilst avoiding the effects of aliasing, improving zero-shot testing performance over alternative continuous formulations, such as SGConv, which don't have any anti-aliasing guarantees. It is important to note that dilated kernels and sparse kernels are not continuous parameterizations and are therefore not suitable models for performing zero-shot changes in input resolution.

### 5.4  ImageNet Classification

To evaluate MRConv on a large-scale task, we employ the ImageNet classification benchmark [41], which consists of 1.28 million high-resolution training images of size 224×224 and 1000 classes. As a base architecture, we choose ConvNeXt [32], a fully convolutional model that enhances the ResNet architecture by incorporating elements from Vision Transformers. To assess `MRConv`, we

| Model (Input length) | sCIFAR (1024) |
|---|---|
| Transformer | 62.2 |
| *State Space Models:* | |
| S4D | 89.92 |
| S5 | 90.10 |
| S4 | 91.80 |
| Liquid-S4 | 92.02 |
| *Convolutional Models:* | |
| CKConv | 63.74 |
| TrellisNet | 73.42 |
| FlexConv | 80.82 |
| Long Conv, Geom Init | 92.1 |
| Long Conv, Learnable Butterfy | 92.5 |
| CCNN | 93.08 |
| MultiresNet | 93.15 |
| MRConv, Dilated | **94.26** |
| MRConv, Fourier | 92.67 |
| MRConv, Fourier+Sparse | 92.40 |

(a) sCIFAR

| Model (Input length) | 16 kHz (16,000) | 8 kHz (8,000) |
|---|---|---|
| *CNNs:* | | |
| InceptionNet | 61.24 | 05.18 |
| ResNet-1 | 77.86 | 08.74 |
| XResNet-50 | 83.01 | 07.72 |
| ConvNet | 95.51 | 07.26 |
| *State Space Models:* | | |
| S4D-LegS | 95.83 | 91.08 |
| S4-LegS | 96.08 | 91.32 |
| Liquid-S4 | 96.78 | 90.00 |
| S5 | 96.52 | 94.53 |
| *Convolutional Models:* | | |
| SGConv | 96.42 | 94.29 |
| MRConv, Dilated | 96.47 | - |
| MRConv, Fourier | **96.82** | **95.05** |
| MRConv, Fourier+Sparse | 95.21 | - |

(b) Speech Commands

Table 3: **Left**: Test accuracy on sCIFAR pixel-level 1D image classification. **Right**: Test accuracy on 35-way Speech Commands classification task [47]. Each model is trained on one-second 16kHz audio waveforms and then tested at 16kHz and 0-shot at 8kHz.

| Model | 2D Bias | FLOPs | Top 1 Acc. |
|---|---|---|---|
| *ConvNeXt-T* | ✓ | 4.5G | 82.1 |
| SGConvNeXt-T | ✗ | 4.3G | 82.0 |
| MRConvNeXt-T | ✗ | 4.3G | **82.2** |
| *ConvNeXt-S* | ✓ | 8.7G | 83.1 |
| SGConvNeXt-S | ✗ | 8.3G | **83.4** |
| MRConvNeXt-S | ✗ | 8.3G | **83.4** |
| *ConvNeXt-B* | ✓ | 15.4G | 83.8 |
| SGConvNeXt-B | ✗ | 14.6G | **83.9** |
| MRConvNeXt-B | ✗ | 14.6G | **83.9** |

(a) ImageNet Classification

| Model | Avg. | Speed Text | Image |
|---|---|---|---|
| S4 | 86.09 | 1× | 1× |
| MEGA-Chunk ($\mathcal{O}(L)$) | 85.66 | 0.3× | 0.7× |
| MRConv-*B* | 87.77 | 0.4× | 0.6× |
| MRConv-*B* (Rep) | **87.77** | 1.5× | 1.3× |
| MEGA ($\mathcal{O}(L^2)$) | **88.21** | 0.1× | 0.5× |
| MRConv-*L* | 88.20 | 0.2× | 0.3× |
| MRConv-*L* (Rep) | 88.20 | 0.9× | 0.7× |

(b) LRA Compute Comparison

Table 4: **Left**: Top-1 test accuracy on ImageNet classification [41]. **Right**: Inference time speed comparison between *Base* and *Large* versions of MRConv and linear and quadratic attention versions of *MEGA* [34]. We denote *Rep* as reparameterized models. By scaling MRConv up with more parameters we match the performance of MEGA with quadratic attention, whilst also being more efficient.

replace the standard 7x7 2D convolutional layers in each block with 1D MRConv blocks, flattening the 2D features in each layer to 1D sequences. We denote our model MRConvNeXt and use the same hyperparameter settings for each Tiny/Small/Base model from ConvNeXt without any changes.

Comparing MRConvNeXt with ConvNeXt and SGConvNeXt [29], another 1D convolution model, MRConvNeXt equipped with Fourier + Sparse kernels achieves SoTA performance, outperforming ConvNeXt at every model size. As highlighted by [29], 1D FFT convolutions use fewer FLOPs than standard convolutions, although empirically, the throughput decreases. Using optimized CUDA kernels for 1D FFT convolutions, we close the gap between theoretical and empirical throughput as shown in Figure 3a, comfortably outperforming Swin [31] a powerful vision transformer, and improving the throughput-accuracy frontier when compared to standard ConvNeXt.

# 6 Discussion & Conclusion

In this work, we introduced `MRConv`, a simple yet effective method for parameterizing long-convolution kernels. We develop three kernel parameterizations and demonstrate through experimentation how each approach is suited to different data modalities. Further, we show the importance of having a learnable decay due to differing model characteristics with depth. Finally, we highlight `MRConv`'s leading performance on LRA, sCIFAR, Speech Commands and ImageNet classification.

However, our model is not without its limitations. Training with parallel branches requires computing many more convolutions, increasing memory usage and slowing down training. Currently, parallel training is necessary due to the presence of batch normalization, which is non-linear during training. For future work, we aim to remove batch normalization, potentially through initialization [8] or linear rescaling (Equation 9), allowing for reparameterization during training, significantly reducing training costs. Our model also lacks input dependency. Whilst this does not affect performance on natural data, such as images and audio, on discrete information-dense sequences, such as text, linear-time transformers still outperform `MRConv` (see Table 1). For future work, we propose introducing input dependency into our model, using either Hyena recurrences [38] or by combining with self-attention similar to MEGA [34]. Finally, unlike SSMs, our model doesn't support fast autoregressive inference by construction. However, we note an equivalence between kernels constructed as the sum of Fourier basis functions and SSMs has already been established [24]. We propose converting `MRConv` equipped with Fourier kernels into a multi-resolution SSM as future work.

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

# Appendix

## A  Additional Background

We provide some additional background information that was not included in the main body of the paper.

### A.1  Structural Reparameterization

It has been shown that training multi-branch convolutional layers with different combinations of paths, scales and complexities can enrich the feature space, improving performance over a single convolutional layer [9–14, 26]. However, such architectures lead to increased training and inference costs. Developed in the vision community, *structural reparameterization* exploits the ability to merge multi-branch convolutional architectures during training into a single convolution at inference by applying a set of equivalent transformations to the parameters of the convolutional layers. In general, structural reparameterization trades off additional training costs for performance.

A key component of structural reparameterization is the use of BatchNorms (BNs),

$$\mathrm{BN}(x) = \frac{x - \mu}{\sqrt{\sigma + \epsilon}} * \gamma + \beta \tag{17}$$

where $\mu$ is the accumulated mean, $\sigma$ the standard deviation, $\gamma$ the learned scaling factor and $\beta$ the learned bias. BNs provide training time non-linearity, causing the kernels to undergo different optimization dynamics than training an equivalent reparameterized convolution. After training BN parameters can be merged with the the preceding convolution layer parameters as,

$$\overline{W} = \frac{\gamma}{\sigma} W, \qquad \overline{b} = -\frac{\mu\gamma}{\sigma} + \beta \tag{18}$$

Recently, structural reparameterization has been used to train large kernels alongside a small kernel, forming an implicit ensemble of models with differing receptive fields, successfully increasing the size of the receptive field without losing the ability to capture small-scale dependencies between inputs [30, 14, 13].

# B  Additional Methods

Here we discuss some additional methods that we experimented with but did not include in the main body of the paper.

## B.1  Multi-Head Convolutions

Similar to multi-head attention, SSMs and prior long-convolution methods generate a *multi-head structure* by applying independent long convolutions to separate copies of the input. Each copy of the input is termed a *head*.

**Remark.** *For an input $u \in \mathbb{R}^{D \times L}$, an multi-head convolution with $H$ heads processes the input $H$ times and has $HD$ number of independent depthwise convolutions.*

## B.2  Reduced-Dimension Convolutions

It has been shown that during training long-convolutions converge to equivalent low-dimensional SSMs and has therefore proven beneficial to apply independent long convolutions over *groups of channels* instead of each channel individually [35]. This reduces the dimensionality of the convolution requiring fewer convolutional filters. Given an input $u \in \mathbb{R}^{D \times L}$, we apply *reduced-dimension* convolutions by:

1. Splitting the dimension $D$ into $G$ groups of size $M = D/G$ such that $u \in \mathbb{R}^{G \times M \times L}$.
2. Applying a set of $M$ depthwise separable convolutions with kernel $k \in \mathbb{R}^{M \times L}$ independently over $G \times M$.
3. Concatenating the outputs $y \in \mathbb{R}^{G \times M \times L}$ over the groups $G$ such that $y \in \mathbb{R}^{D \times L}$.

**Remark.** *For an input $u \in \mathbb{R}^{D \times L}$, a reduced-dimension convolution with $G$ groups reduces the number of independent depthwise convolutions to $D/G$.*

## B.3  Connections to SSMs

We establish a relationship between our Fourier parameterised kernels and SSMs. Let $k \in \mathbb{R}^L$ be a real-valued kernel of length $L$, sampled at $1/L$ Hz and parameterized in the Fourier domain as,

$$k[t] = \frac{1}{L} \sum_{k=0}^{L-1} \hat{c}_k \exp\left(2\pi i \frac{tk}{L}\right) \tag{19}$$

where $\hat{c}_k \in \mathbb{C}$ are complex valued parameters and $\hat{c}_k = \hat{c}^*_{-k}$ such that kernel values $k$ are real.

From [24] we can represent the above convolution kernel as the following SSM,

$$\boldsymbol{A}_{nk} = \begin{cases} -2 & n = k = 0 \\ -2\sqrt{2} & n = 0, k \text{ odd} \\ -2\sqrt{2} & k = 0, n \text{ odd} \\ -4 & n, k \text{ odd} \\ 2\pi k & n - k = 1, k \text{ odd} \\ -2\pi n & k - n = 1, n \text{ odd} \\ 0 & \text{otherwise} \end{cases} \qquad \boldsymbol{B}_n = \begin{cases} 2 & n = 0 \\ 2\sqrt{2} & n \text{ odd} \\ 0 & \text{otherwise} \end{cases} \tag{20}$$

**Theorem ([24] 6.).** *As state size $N \to \infty$, the SSM in Eq 20 is a time-invariant orthogonal state space model defined by the truncated Fourier basis functions, orthonormal on $[0, 1]$, $\{p_n\}_{n \geq 0} = [1, c_0(t), s_0(t), \cdots]$, where $c_m(t) = \sqrt{2}\cos(2\pi mt)$ and $s_m(t) = \sqrt{2}\sin(2\pi mt)$ for $m = 0, \cdots, N/2$.*

Hence, as the state size $N \to \infty$, the corresponding convolution kernel to the above SSM is a linear combination of truncated Fourier basis functions with support on the unit interval $[0, 1]$, controlled by the vector of coefficients $C$ whose parameters correspond to $\hat{c}$ in our explicit formulation in Eq 19. It is interesting to note that whilst we exploit the FFT to generate our convolution in $\mathcal{O}(L \log L)$, to generate the equivalent kernel given the recurrent form of the SSM takes $\mathcal{O}((L + D) \log^2(L + D))$ time using Cauchy and Vandermonde matrix multiplications, which, even with efficient CUDA implementations, results in considerably slower performance than highly optimized FFTs (see Table 2).

# C  Implementation Details

We provide algorithmic implementations of both the dilated and Fourier kernel parameterizations and the aggregation step in `MRConv`.

## C.1  Kernel Parameterization

---

**Algorithm 1** MRConv, Dilated

---

1: **Input:** Input sequence $\boldsymbol{u} : [B, D, L]$, depth $N$
2: **Output:** $\tilde{\boldsymbol{c}} : [B, D, L, N]$
3: **for** $i = 1$ **to** $N$ **do**
4:    $\boldsymbol{h}_i \leftarrow \text{ImplicitFilter}(D, i)$                                  $// [D, 2^i]$
5:    $\boldsymbol{c}_i \leftarrow \text{DepthwiseCausalConv}(\boldsymbol{u}, \boldsymbol{h}_i)$          $// [B, D, L, 1]$
6:    $\tilde{\boldsymbol{c}}_i \leftarrow \text{BN}_i(\boldsymbol{c}_i)$                                  $// [B, D, L, 1]$
7: **end for**
8: $\tilde{\boldsymbol{c}} = \text{Concat}(\{\tilde{\boldsymbol{c}}_i\}_{i=1}^N)$

---

**Algorithm 2** MRConv, Fourier

---

1: **Input:** Input sequence $\boldsymbol{u} : [B, D, L]$, depth $N$, Complex Kernels $\hat{\boldsymbol{k}} : [D, L]$
2: **Output:** $\boldsymbol{c} : [B, D, L, N]$
3: **for** $i = 1$ **to** $N$ **do**
4:    $\boldsymbol{h}_i \leftarrow \text{IFFT}[\text{ZeroPad}(\hat{\boldsymbol{k}}_i)]$
5:    $\boldsymbol{c}_i \leftarrow \text{DepthwiseCausalConv}(\boldsymbol{u}, \boldsymbol{h}_i)$          $// [B, D, L, 1]$
6:    $\tilde{\boldsymbol{c}}_i \leftarrow \text{BN}_i(\boldsymbol{c}_i)$                                  $// [B, D, L, 1]$
7: **end for**
8: $\tilde{\boldsymbol{c}} = \text{Concat}(\{\tilde{\boldsymbol{c}}_i\}_{i=1}^N)$

---

## C.2  Multi-Resolution Convolutions

---

**Algorithm 3** MRConv, Aggregation

---

1: **Input:** Input sequence $\boldsymbol{u} : [B, D, L]$, weights $\boldsymbol{\alpha} : [N, D]$, depth $N$.
2: **Output:** $\boldsymbol{y} : [B, D, L]$
3: $\tilde{\boldsymbol{c}} \leftarrow \text{MultiResConv}(\boldsymbol{u}, N)$                                  $// [B, D, L, N]$
4: $\boldsymbol{y} \leftarrow \boldsymbol{\alpha}^T \boldsymbol{c}$                                  $// [B, D, L]$

---

# D    Experiments and Configurations

In this section we provide additional experimental details, including model architecture, hyperparameters and task details. We also include some further ablation studies not included in the main body of the paper.

## D.1    Runtime Comparison

To evaluate the computational efficiency of MRConv we benchmark its runtime against multi-head attention, FlashAttention [7], and S4D [23]. The evaluation was conducted on an NVIDIA A100-40GB GPU where each model has a hidden dimension of 768 and we use a batch size of 64. Both FlashAttention and MRConv, which uses FlashFFTConv [18], utilize bfloat16 precision. For MRConv we use an initial kernel size of size 128. The runtime was computed as an average over 1000 forward passes. We get out of memory errors for MHA at sequence length $\geq 4096$.

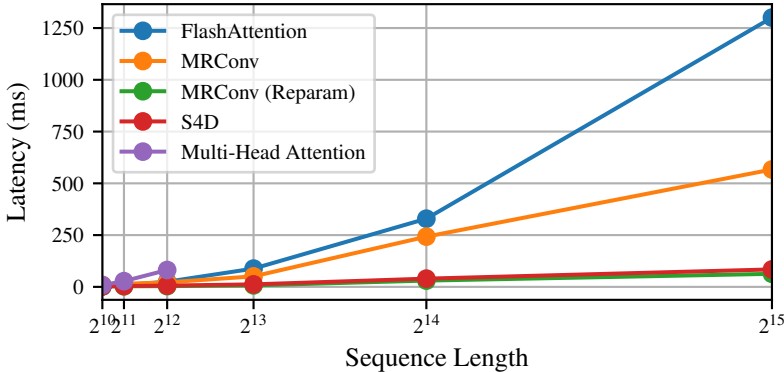

Figure 4: **Runtime Comparison.** Runtime comparison of MRConv versus PyTorch's Multi-Head attention (MHA) implementation and FlashAttention at inference with increasing sequence length.

## D.2 Default Hyperparameters

**Base Models** Table 5 presents the highest performing hyperparameters for each base model used for each experiment. For all experiments we ensure that the *total number of trainable parameters* stays comparable with baseline methods.

| Dataset | Kernel Type | Depth | Features | Kernel Size | Bidirectional | Norm | Prenorm | Dropout | Kernel LR | LR | WD | Batch Size | Epochs |
|---|---|---|---|---|---|---|---|---|---|---|---|---|---|
| ListOps | Fourier | 8 | 128 | 2 | ✗ | BN | ✗ | 0.05 | 0.001 | 0.003 | 0.05 | 50 | 40 |
| Text | Fourier | 6 | 256 | 1 | ✗ | BN | ✓ | 0.05 | 0.001 | 0.005 | 0.05 | 16 | 32 |
| Retrieval | Fourier | 6 | 256 | 1 | ✗ | BN | ✓ | 0.05 | 0.001 | 0.003 | 0.05 | 64 | 20 |
| Image | Dilated | 6 | 512 | 8 | ✗ | LN | ✗ | 0.1 | 0.001 | 0.0045 | 0.05 | 50 | 200 |
| Pathfinder | Fourier+Sparse | 6 | 256 | 16 | ✓ | BN | ✓ | 0.1 | 0.001 | 0.005 | 0.03 | 64 | 200 |
| Path-X | Fourier | 6 | 256 | 64 | ✓ | BN | ✓ | 0 | 0.001 | 0.004 | 0.03 | 16 | 50 |
| sCIFAR | Dilated | 10 | 512 | 8 | ✗ | LN | ✗ | 0.2 | 0.001 | 0.0045 | 0.05 | 50 | 300 |
| SC | Fourier | 6 | 128 | 32 | ✓ | BN | ✓ | 0.1 | 0.001 | 0.005 | 0.05 | 16 | 40 |

Table 5: Experiment hyperparameters for `MRConv`-*Base* variants

**Large Models** Table 6 presents the highest performing hyperparameters for each large model used in LRA. For this set of experiments we ensure that the *computational resources* (throughput and memory requirements) at inference are comparable with baseline methods that use quadratic attention.

| Dataset | Kernel Type | Depth | Features | Kernel Size | Bidirectional | Norm | Prenorm | Dropout | Kernel LR | LR | WD | Batch Size | Epochs |
|---|---|---|---|---|---|---|---|---|---|---|---|---|---|
| ListOps | Fourier | 16 | 128 | 1 | ✗ | BN | ✗ | 0.05 | 0.001 | 0.003 | 0.05 | 50 | 40 |
| Text | Fourier+Sparse | 6 | 384 | 1 | ✗ | BN | ✓ | 0.1 | 0.001 | 0.005 | 0.05 | 16 | 32 |
| Retrieval | Fourier | 6 | 384 | 1 | ✗ | BN | ✓ | 0 | 0.001 | 0.003 | 0.05 | 64 | 20 |
| Image | Dilated | 10 | 512 | 8 | ✗ | LN | ✗ | 0.2 | 0.001 | 0.0045 | 0.05 | 50 | 200 |
| Pathfinder | Fourier+Sparse | 12 | 256 | 32 | ✓ | BN | ✓ | 0.05 | 0.001 | 0.005 | 0.03 | 64 | 200 |
| Path-X | Fourier | 12 | 256 | 64 | ✓ | BN | ✓ | 0 | 0.001 | 0.004 | 0.03 | 16 | 50 |

Table 6: Experiment hyperparameters for `MRConv`-*Large* variants

### D.2.1 Optimization

We follow the optimization approach presented in [23], which uses the AdamW optimizer with a global learning rate and weight decay and a separate smaller learning rate with no weight decay specifically for the kernel parameters. All experiments use a cosine annealing learning rate schedule with linear warmup.

### D.2.2 Compute Infrastructure

All LRA, sCIFAR and Speech Commands experiments were run using a single 40GB A100 GPU apart from Retrieval and Path-X and Speech Commands where we use 2 40GB A100s.

## D.3 Long Range Arena

Here we provide context and implementation details for each of the LRA tasks. In our work we use the now standard pre-processing steps from [22].

1. **ListOps**: The dataset consists of sequences of nested mathematical operations, including brackets, lists of numbers and operators `min`, `max`, `median` and `summod`. In total, there are 17 unique token values, including possible integers, which are encoded as one-hot vectors. The task is to compute the integer result of the operations, e.g. $[\min 7\ 4\ [\max 2\ 5\ ]\ 3\ 4\ [\text{median}\ 0\ 1\ 6\ 9\ ,\ 2\ ]] \rightarrow 1$. This is described as a 10-class classification problem where each class represents an integer result of the expression. The sequences are of unequal length and shorter sequences within a batch are zero-padded up to a maximum length of $2,000$. There are $96,000$ training examples, $2,000$ validation examples and $2,000$ test sequences.

2. **Text**: The task is binary text classification on whether a movie review is positive or negative. The dataset is constructed from the IMDB reviews dataset. Each review is encoded on the character level as one-hot vectors, with 129 possible values. Encoding on the character level naturally simulates a longer input sequence. Sequences are of unequal length and are either zero-padded or truncated to a maximum length of $4,000$. There are $25,000$ training examples, no validation examples and $25,000$ test examples.

3. **Retrieval**: The task is to identify if two papers share a citation link. The dataset is based on the ACL Anthology Network dataset. The input is constructed by concatenating each document and compressing each sequence individually, before passing the compressed representations to a linear classifier. Similar to the text classification setup, text inputs are encoded on the character level as one-hot vectors with 97 unique values. Input sequences of unequal length are either zero-padded or truncated to a maximum length of $4,000$, which after concatenation makes the total sequence length $8,000$. There are $147,086$ training pairs, $18,090$ validation pairs and $17,437$ test pairs.

4. **Image**: The task is image classification using the CIFAR-10 dataset, flattened into a 1D sequence of length 1024 and grayscaled. Each pixel value is normalized to have zero mean and unit variance. The dataset consists of ten classes. There are $45,000$ training examples, $5,000$ validation examples and $10,000$ test examples.

5. **Pathfinder**: The dataset consists of $32 \times 32$ grayscale images which contain 2 small circles and several dashed lines. The images are then flattened to a 1D sequence of length $1,024$ and normalized to be in the range $[-1, 1]$. The task is to identify whether the circles are connected by one of the dashed lines or not, formulated as a binary classification problem. There are $160,000$ training examples, $20,000$ validation examples and $20,000$ test examples.

6. **Path-X**: The task is a harder version of the pathfinder challenge, where the input images are $128 \times 128$, which when flattened form a sequence of length $16,384$, and there are more distraction lines. This makes the Path-X challenge significantly harder than Pathfinder.

### D.3.1 Hyperparameter Sweeps

For all experiments we performed large hyperparameter sweeps on the following parameters:

- Kernel size: [1, 2, 4, 8, 16, 32]
- Learning Rate: [0.03, 0.05, 0.07]
- Dropout: [0, 0.1, 0.2, 0.3]

### D.3.2 Extended LRA Results

We provide further LRA results that include quadratic attention Transformers and citations in Table 7.

### D.3.3 Path-X Implementation Details

We found that due to the extreme length of the input sequences on Path-X, $(16, 384)$, `MRConv` required a large initial kernel size to ensure the longer kernels had a large enough rank such that they wouldn't just collapse to a bias term. However, using large initial kernel sizes also led to overfitting on local information, and we didn't perform better than random guessing. Additionally, We observed that

| Model (Input length) | ListOps (2,048) | Text (4,096) | Retrieval (4,000) | Image (1,024) | Pathfinder (1,024) | Path-X (16,384) | Avg. |
|---|---|---|---|---|---|---|---|
| *Quadratic-Time Transformers:* | | | | | | | |
| Transformer [46] | 36.37 | 64.27 | 57.46 | 42.44 | 71.40 | ✗ | 53.66 |
| Transformer + SPT [1] | 59.15 | 88.81 | 90.38 | 76.00 | 88.49 | 88.05 | 81.81 |
| MEGA [34] | **63.14** | 90.43 | 91.25 | 90.44 | 96.01 | 97.98 | **88.21** |
| *Linear-Time Transformers:* | | | | | | | |
| BST [16] | 61.49 | 87.63 | 90.51 | **91.07** | 95.75 | 95.28 | 86.96 |
| SPADE-Chunk [49] | 60.50 | **90.69** | 91.17 | 88.22 | 96.23 | 97.60 | 87.40 |
| MEGA-Chunk [34] | 58.76 | 90.19 | 90.97 | 85.80 | 94.41 | 93.81 | 85.66 |
| *State Space Models:* | | | | | | | |
| S4D-LegS [23] | 60.47 | 86.18 | 89.46 | 88.19 | 93.06 | 91.95 | 84.89 |
| S4-LegS [22] | 59.60 | 86.82 | 90.90 | 88.65 | 94.20 | 96.35 | 86.09 |
| Liquid-S4 [25] | 62.75 | 89.02 | 91.20 | 89.50 | 94.8 | 96.66 | 87.32 |
| S5 [43] | 62.15 | 89.31 | 91.40 | 88.00 | 95.33 | 98.58 | 87.46 |
| *Convolutional Models:* | | | | | | | |
| CCNN [40] | 43.60 | 84.08 | - | 88.90 | 91.51 | ✗ | - |
| Long Conv [17] | 62.2 | 89.6 | 91.3 | 87.0 | 93.2 | 96.0 | 86.6 |
| SGConv [29] | 61.45 | 89.20 | 91.11 | 87.97 | 95.46 | 97.83 | 87.17 |
| *Ablations:* | | | | | | | |
| MRConv-*B*, Dilated | 60.90 | 86.38 | 88.30 | 90.37 | 94.42 | ✗ | 78.40 |
| MRConv-*B*, Fourier | 62.40 | 89.26 | 91.44 | 88.55 | 95.03 | 97.82 | 87.42 |
| MRConv-*B*, Fourier+Sparse | 62.10 | 89.26 | 91.35 | 89.07 | 95.55 | ✗ | 79.56 |
| MRConv-*L*, Dilated | 61.25 | 88.36 | 89.78 | 90.55 | 95.22 | ✗ | 79.19 |
| MRConv-*L*, Fourier | 62.45 | 89.40 | **91.48** | 89.30 | 95.75 | **98.65** | 87.84 |
| MRConv-*L*, Fourier+Sparse | 61.65 | 89.42 | 91.35 | 89.15 | **96.64** | ✗ | 79.70 |
| *Ours:* | | | | | | | |
| MRConv-*B* | 62.40 | 89.26 | 91.44 | 90.37 | 95.55 | 97.82 | 87.81 |
| MRConv-*L* | 62.45 | 89.42 | **91.48** | 90.55 | **96.64** | **98.65** | 88.20 |

Table 7: **Test accuracy on the Long Range Arena Benchmarks [44]**. We follow the standard training procedures introduced in [23]. Bold scores indicate the highest performing model on a given task and underlined the second best performing. ✗ indicates a model did not do better than random guessing and - that a result was not available. In this table we only include results from other non-input-dependent models.

learning the linear combination of each sub-kernel resulted in some overfitting when compared to using a fixed kernel decay.

To address the training instabilities, we simplified our reparameterization scheme by concatenating kernels together instead of summing them as is in SGConv, and using a fixed kernel decay rate throughout. These changes helped to stabilize the training process and improved generalization, as shown in Table 8.

| Model modification | Path-X | |
|---|---|---|
| | Accuracy | Change |
| MRConv-*B*, Fourier | 50.00 | - |
| + Concatenate sub-kernels | 96.19 | + 46.19 |
| + Fixed kernel decay | **97.83** | + 47.83 |

Table 8: **Path-X design ablations.** We find that concatenating kernels together instead of summing dramatically improved performance on Path-X from random guessing (50%) to 96.19% accuracy and using a fixed kernel deacy reduced overfitting, further improving performance to 97.83%.

### D.3.4 Convergence Plots

We provide additional training plots corresponding to the ablations reported in Table 2. In particular, the convergence plots highlight the difficulty in training long dense convolution kernels which overfit to the training data, as seen in Figure 5.

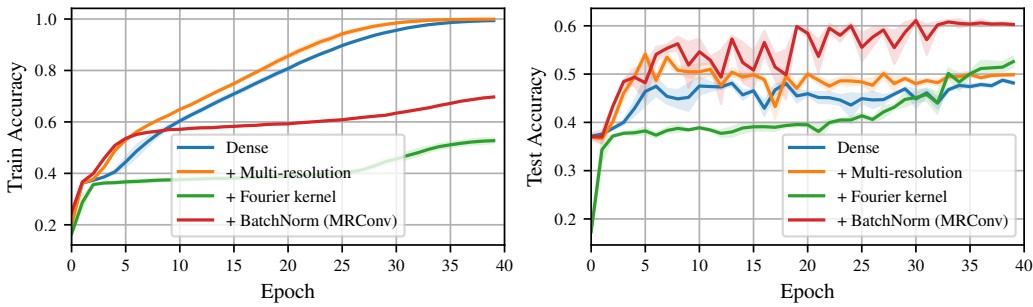

Figure 5: **Convergence Plots on** `ListOps`. Corresponds to ablations in Table 2

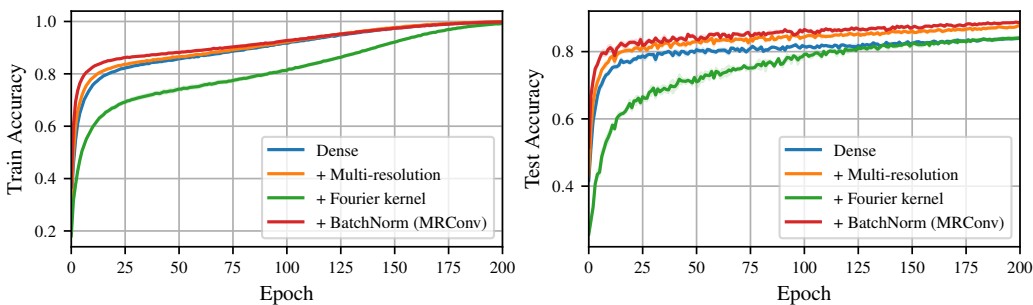

Figure 6: **Convergence Plots on** `sCIFAR`. Corresponds to ablations in Table 2

### D.3.5   Initial Resolution Ablation

We perform an additional ablation study on the effects of different initial kernel sizes on the `ListOps` and `Image` datasets from LRA. Table 9 reports our results, highlighting how kernel size can have an effect of final accuracy, with shorter kernel sizes favoring more discrete data types such as found in the `ListOps` task and longer kernel sizes smoother data such as flattened images used in the `sCIFAR` task.

| Kernel Type | ListOps | | | Image | | |
|---|---|---|---|---|---|---|
| | $l_0$ | $N$ | Accuracy | $l_0$ | $N$ | Accuracy |
| Fourier Kernel | 1 | 11 | 61.80 | 8 | 8 | 86.69 |
| | 2 | 10 | 62.40 | 16 | 7 | 88.39 |
| | 4 | 9 | 61.10 | 32 | 6 | 88.55 |

Table 9: **Initial kernel size albations**. Effect of the initial kernel size $l_0$ on ListOps and Image tasks from LRA. We note that the initial kernel size also effects the number of independent resolutions $N = \log_2(L/l_0) + 1$ which we include in the table for clarity.

### D.3.6   LayerNorm Ablation

Whilst LayerNorm cannot be used in conjunction with structural parametrization since it remains a non-linear operation during inference, we provide a further ablation where we substitute BatchNorm for LayerNorm. Our results in Table 10 show that the performance of LayerNorm is comparable to BatchNorm. This further emphasises the suitability of BatchNorm for structural reparameterization, on top of being able to reparameterize BatchNorm at inference.

### D.3.7   MLP Kernels

Further to the Dilated, Sparse and Fourier kernel parameterizations suggested introduced in Section 3.3, we also include experimental results performed using multi-resolution kernels implicitly parameterized by an MLP of fixed size. Kernels parameterized by MLPs have previsouly shown to be

| Norm Type | ListOps | Image |
|-----------|---------|-------|
| BatchNorm | 62.40 | 89.30 |
| LayerNorm | 60.10 | 87.80 |

Table 10: **Normalization albations**. Effect of different normalization layers. We find that there is very little difference between using LayerNorm of BatchNorm but emphasize the benefit of using BatchNorm which enables structural reparameterization.

successful in sequence modelling tasks [40, 39, 38]. Our results in Table 11 show however that MLP based kernels when placed within our multi-resolution framework underperform both dilated and Fourier kernels.

| Kernel Type | ListOps | | Image | |
|-------------|---------|----------|--------|----------|
| | Params | Accuracy | Params | Accuracy |
| Dilated | 759K | 59.25 | 3.5M | 90.37 |
| Fourier | 332K | 62.40 | 3.8M | 88.55 |
| Fourier+Sparse | 420K | 62.25 | 4.0M | 89.07 |
| MLP | 580K | 60.08 | 3.6M | 85.71 |

Table 11: **Kernel parameterization albations**. Performance of different kernel parameterizations on ListOps and Image tasks from LRA.

## D.4 Pixel-Level 1D Image Classification

The task is 10-way image classification using the CIFAR-10 dataset. The task is identical to the LRA Image task except that flattened full colour images are used as input instead. Table 12 reports additional results and citations.

| Model (Input length) | sCIFAR (1024) |
|---|---|
| Transformer [46] | 62.2 |
| *RNNs:* | |
| LSTM [21] | 63.01 |
| LipschitzRNN [15] | 64.2 |
| r-LSTM [45] | 72.2 |
| *State Space Models:* | |
| S4D [23] | 89.92 |
| S5 [43] | 90.10 |
| S4 [22] | 91.80 |
| Liquid-S4 [25] | 92.02 |
| *Convolutional Models:* | |
| CKConv [40] | 63.74 |
| TrellisNet [5] | 73.42 |
| FlexConv [39] | 80.82 |
| Long Conv, Geom Init [17] | 92.1 |
| Long Conv, Learnable Butterfy [17] | 92.5 |
| CCNN [27] | 93.08 |
| MultiresNet [42] | 93.15 |
| MRConv, Dilated | **94.26** |
| MRConv, Fourier | 92.67 |
| MRConv, Fourier+Sparse | 92.40 |

Table 12: **Test accuracy on sCIFAR 1D Image classification**. Citations refer to the original model where the baseline values is taken from otherise additional citations indicates work in which the baseline value is reported.

### D.4.1 Model Depth Ablation

We perform an ablation study comparing models with increasing depth on sCIFAR with dilated convolutions. Table 13, shows that increasing the model depth has a positive impact on model performance.

| Model | Layers | Parameters | sCIFAR |
|---|---|---|---|
| MRConv, Dilated | 6 | 3.4M | 93.57 |
| | 8 | 4.6M | 94.03 |
| | 10 | 5.7M | **94.26** |

Table 13: **Depth ablation study on sCIFAR dataset.** Increasing the number of layers has a positive effect on model performance.

## D.5 Raw Speech Classification

The task is to classify one of 35 words from the dataset of 1s audio recordings, released by [47], sampled at 16kHz. Sequences are all of the same length (16,000). The zero-shot classification task is constructed by temporally downsampling the validation and test datasets using naive decimation, reducing the signal length to 8,000. There are 24,482 training examples, 5,246 validation examples and 5,247 test examples. Following [43], the data is normalized to be zero-mean and have a standard deviation of 0.2. Table 14 reports citations for each of the baseline methods.

| Model (Input length) | 16 kHz (16,000) | 8 kHz (8,000) |
|---|---|---|
| *CNNs:* | | |
| InceptionNet [36] | 61.24 | 05.18 |
| ResNet-18 [36] | 77.86 | 08.74 |
| XResNet-50 [36] | 83.01 | 07.72 |
| ConvNet [36] | 95.51 | 07.26 |
| *State Space Models:* | | |
| S4D-LegS [23] | 95.83 | 91.08 |
| S4-LegS [22] | 96.08 | 91.32 |
| Liquid-S4 [25] | 96.78 | 90.00 |
| S5 [43] | 96.52 | 94.53 |
| *Convolutional Models:* | | |
| SGConv [29] | 96.42 | 94.29 |
| MRConv, Dilated | 96.47 | - |
| MRConv, Fourier | **96.82** | **95.05** |
| MRConv, Fourier+Sparse | 95.21 | - |

Table 14: **Test accuracy on 35-way Speech Commands classification task [47].** Each model is trained on one-second 16kHz audio waveforms and then tested at 16kHz and 0-shot at 8kHz. Results for the baselines are reported in [23].

## D.6 ImageNet Classification

We use the same training settings as in the original ConvNeXt work by Liu et al. [32]. ConvNeXt uses several downsampling layers and hence the feature maps once flattened correspond to sequences of length $[3136, 784, 196, 49]$ at each stage. When setting the initial kernel size, we take inspiration from RepLKNet [14] which reparameterizes a small $5 \times 5$ kernel into a larger one, and hence we ensure our smallest kernel size is $\geq 25$ and use no more than 5 resolutions. Hence, for each stage we used kernel lengths $[196, 49, 28, 28]$. We used $8 \times$ V100 for training. We provide an extended results table below.

| Model | 2D Bias | Params | FLOPs | Throughput (Image/s) | Top 1 Acc. |
|---|---|---|---|---|---|
| ConvNeXt-T [32] | ✓ | 29M | 4.5G | 1306.3 | 82.1 |
| Swin-T [31] | ✓ | 29M | 4.5G | 1077.4 | 81.3 |
| SGConvNeXt-T [29] | ✗ | 29M | 4.3G | 819.7 | 82.0 |
| MRConvNeXt-T, Fourier | ✗ | 30M | 4.3G | 1080.3 | 82.1 |
| MRConvNeXt-T, Fourier + Sparse | ✗ | 32M | 4.3G | 1080.3 | **82.2** |
| ConvNeXt-S [32] | ✓ | 50M | 8.7G | 902.6 | 83.1 |
| Swin-S [31] | ✓ | 50M | 8.7G | 674.5 | 83.0 |
| SGConvNeXt-S [29] | ✗ | 51M | 8.3G | 500.0 | **83.4** |
| MRConvNeXt-S, Fourier | ✗ | 53M | 8.3G | 739.1 | 83.3 |
| MRConvNeXt-S, Fourier + Sparse | ✗ | 57M | 8.3G | 739.1 | **83.4** |
| ConvNeXt-B [32] | ✓ | 89M | 15.4G | 460.9 | 83.8 |
| Swin-B [31] | ✓ | 88M | 15.4G | 470.9 | 83.5 |
| SGConvNeXt-B [29] | ✗ | 90M | 14.6G | 330.4 | **83.9** |
| MRConvNeXt-B, Fourier | ✗ | 93M | 14.6G | 478.7 | 83.8 |
| MRConvNeXt-B, Fourier + Sparse | ✗ | 98M | 14.6G | 478.7 | **83.9** |

Table 15: **Extended ImageNet classification results**. Other than throughput results, which we computed ourselves, baseline results were taken from the cited paper.

### D.6.1 FLOPs Calculation

We provide a further theoretical analysis on the number of FLOPs required to compute a 1D FFT convolution and a regular 2D convolution. In all comparisons, we will use a batch size of 1 and a channel dimension of 1. An FFT convolution involves: i) a forward FFT of the input, ii) an elementwise product between the input and the kernel, and iii) an inverse FFT of the resulting convolution. The total time complexity is $\mathcal{O}(2L \log L + L)$. In practice, we use the real-valued FFT implemented via the Cooley-Tukey algorithm, which uses FLOPs [3]. On the other hand, regular 2D depthwise separable convolutions with a kernel size of $k$ have a complexity of $\mathcal{O}(k^2 HW)$, requiring $2k^2 HW$ FLOPs with 2 FLOPs needed for each multiply-accumulate operation.

Next, we calculate the number of FLOPs to compute a 2D 7x7 convolution, as utilized in ConvNeXt and a global 1D convolution as utilized in MRConv, for a sequence of increasingly larger (flattened) images. Our results show that our 1D convolutions use fewer FLOPs than the comparable 2D convolution used by ConvNeXt. We note that MRConv uses the same number of FLOPs as SGConv [29].

| Conv Type | Conv Size | (16,16) (256) | (32, 32) (1024) | (64, 64) (4096) | (128, 128) (16,384) | (256, 256) (65,536) | (512, 512) (262,144) |
|---|---|---|---|---|---|---|---|
| 2D | $7 \times 7$ | 25.1K | 100K | 401K | 1.61M | 6.42M | 25.7M |
| 1D | $7 \times 7$ | 8.7K | 43.0K | 205K | 0.95M | 4.32M | 19.4M |

Table 16: **Theoretical FLOPs**. Comparison of theoretical FLOPs between 2D convolutions and 1D convolutions implemented using the FFT. We display both the 2D image size and the length of the equivalent flattened 1D image.

### D.6.2 Throughput

The throughput is measured as the number of images processed per second on a single 24GB 3090 GPU, by using the largest batch size that could fit in memory for a given model and averaging over 50,000 test images.

We include additional results comparing the throughput of `MRConvNeXt` using both PyTorch implemented FFT convolutions and optimized CUDA FFT convolutions from [17]. Use of optimized CUDA kernels results in a 41.5% throughput increase on average. We also include an enlarged version of Figure 3a which plots Top-1 accuracy against throughput.

| Model | FFT Conv. Imp. | Throughput (Image/s) |
|---|---|---|
| MRConvNeXt-T | PyTorch | 819.7 |
|  | CUDA | 1080.3 |
| MRConvNeXt-S | PyTorch | 500.0 |
|  | CUDA | 739.1 |
| MRConvNeXt-B | PyTorch | 330.4 |
|  | CUDA | 478.7 |

Table 17: **ImageNet classification throughput comparison.** We compare the throughput of `MRConvNeXt` when using PyTorch implemented FFT convolutions and fast CUDA implemented FFT convolutions [17]

### D.6.3 Visualization of `MRConvNeXt`-T kernels

In Figure 7 we provide visualization of the reparameterized convolution kernels from $\texttt{MRConvNeXt}-T$ for both Fourier and Fourier + Sparse parameterizations.

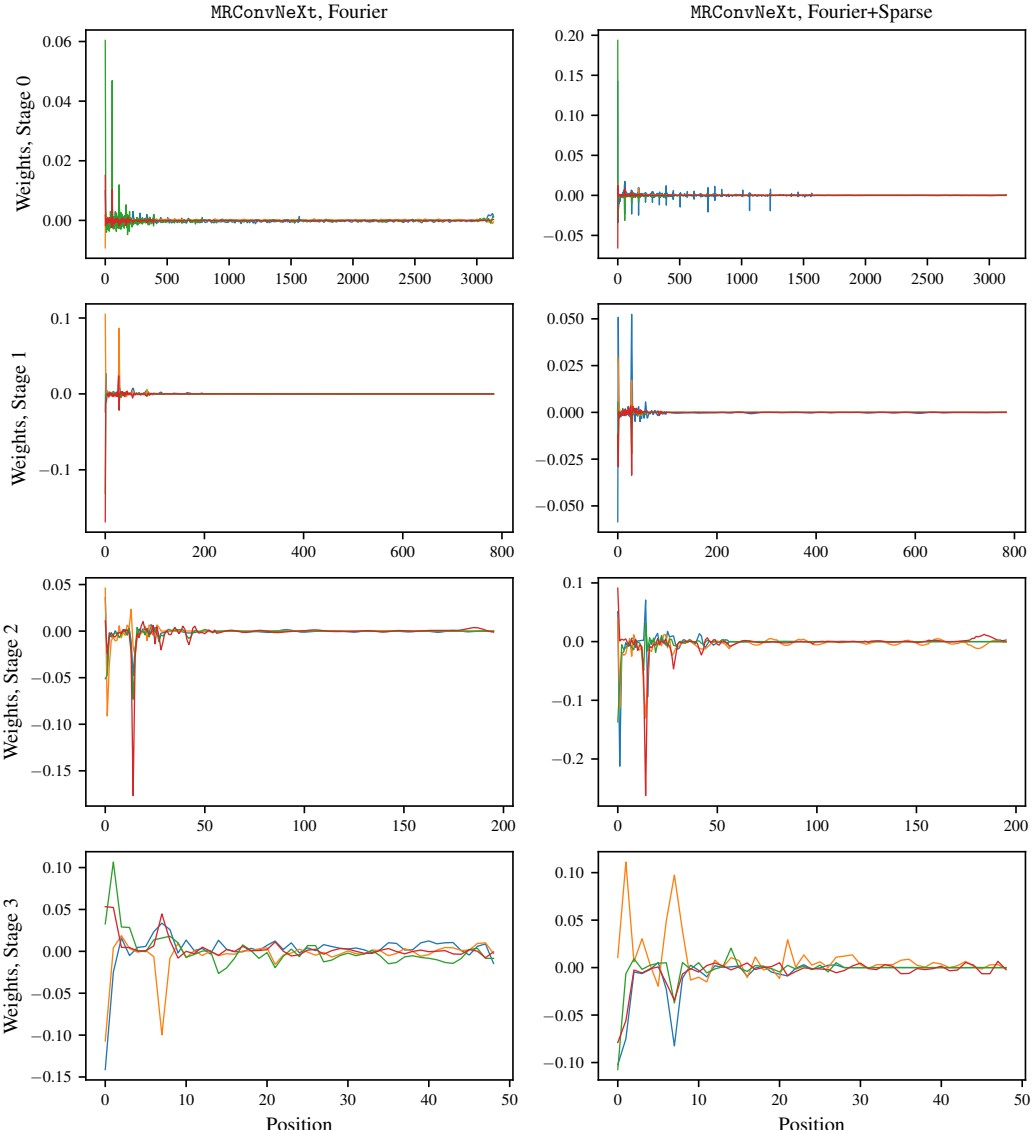

Figure 7: Visualization of learned Kernels from `MRConvNeXt` at different stages for both Fourier and Fourier + Sparse parameterizations.

