# OpenReview forum: "Reparameterized Multi-Resolution Convolutions for Long Sequence Modelling"
_NeurIPS.cc/2024/Conference — NeurIPS 2024 poster_

### Official Review · Reviewer_CE6k · 2024-06-14

**Soundness:** 3
**Presentation:** 3
**Contribution:** 3
**Rating:** 7
**Confidence:** 3

**Summary:**

This paper develops a novel solution to address long-sequence tasks by parameterizing global convolutional kernels. In addition, the paper presents a simple idea, yet it achieves excellent results. Moreover, this paper's clear presentation makes it straightforward to understand. The baselines mentioned in the evaluation section are detailed.

**Strengths:**

1. The problem studied in this paper is very important. Recently, S4 and Mamba have been developed for addressing long sequence problems.
2. The writing of this paper is great and it is easy for readers to follow the entire paper.
3. The evaluation in this paper is thorough and detailed.

**Weaknesses:**

To be honest, I think this paper is really excellent. The only thing I want to see in the rebuttal is the inference time of each model. I am wondering whether the authors can show some experimental results about the inference time of each model.

**Questions:**

See weaknesses.

**Limitations:**

I think the authors have adequately addressed the limitations.

---

> ### Author Rebuttal · Authors · 2024-08-06
>
> We thank the reviewer for supporting the acceptance of our paper and appreciating our thorough evaluation and analysis. We address the reviewer's only question below:
>
> ## Runtime Comparison
> Please see our Author's rebuttal at the top of the page and the attached pdf. Our results show that MRConv is substantially faster than the efficient FlashAttention implementation, particularly for long sequences. Furthermore, our results emphasize that even our non-reparameterized kernels are efficient and remain computationally faster than FlashAttention, aligning with our theoretical complexity calculations.

---

### Official Review · Reviewer_JNPq · 2024-07-12

**Soundness:** 3
**Presentation:** 3
**Contribution:** 3
**Rating:** 6
**Confidence:** 5

**Summary:**

The paper presents MRconv, a novel type of global convolution layer designed for long 1-D sequence modeling. MRconv is built on an efficient and effective parameterization that produces normalized multi-resolution convolutions. The authors conduct comprehensive empirical evaluations on several benchmarks, including ImageNet-1K, LRA, sCIFAR, and Speech Commands, demonstrating that MRconv achieves near SoTA performance on several tasks and modalities. Finally, several ablations are conducted to justify the design choices and explore additional variants of the layer.

**Strengths:**

1. The authors demonstrate a **comprehensive empirical evaluation** of the method across several benchmarks and domains, achieving near SOTA results on real-world tasks such as image and small-scale speech classification, highlighting the robustness and versatility of the approach.

2. The **novel and efficient** parametrization, particularly advantageous during inference, yields promising results, as evidenced by Figure 3 (Left) and Table 4.

3. **Simplicity:** The layer is relatively simple and can be explained with just a few equations, making it easier for the community to adopt (as described in Algorithms 1,2,3).

**Weaknesses:**

1. **Empirical evaluation should be improved:**

- 1.a. **Results on NLP**: Global conv layers such as S4, Hyena, and Gated state-space are effective in language modeling. Thus, assessing the language modeling capabilities of MRconv (including both positive or negative results), can enhance the manuscript. Since MRconv is implemented within the S4 repository (which includes WikiText-103), conducting such experiments requires minimal additional effort.

- 1.b. Conducting experiments with **small-scale synthetic tasks** and controlled environments can provide valuable insights into the properties, weaknesses, and strengths of the layer compared to alternatives. Examples of such tasks include atomic tasks [1], copy, selective copy [2], associative recall [3], and others [4]. Even negative results are important, as they offer valuable information about the limitations and failure cases of the layer.

- 1.c.  **Gated convolutions** have proven to be highly effective in various domains. Understanding their critical role in each domain and how much they can enhance MRconv is both important and informative. This knowledge can improve the usability of these layers.

2. **Efficiency benchmarking should be improved:**

- 2.a. **FLOPS comparison:** The provided comparison of FLOPs focuses solely on ImageNet, utilizing the ConvNeXt backbone. However, this comparison may be less informative as most of the FLOPs in these backbones are not located in the tested layers (SGconv, Conv2D, MRconv). Instead, the majority of the FLOPs are found in the MLP (or 1x1 Conv) layers. Therefore, I find these comparisons less relevant. Am I overlooking something here? If not, please perform a FLOPs comparison in other, more relevant regimes. Additionally, I recommend the authors include a figure showing the FLOPs (and/or latency, and throughput) on the y-axis for various sequence lengths (x-axis) across several layers (MRconv, SGconv, SSM variant, Conv1D, and others) to fully describe the empirical complexity of the method.

- 2.b. **A normalized amount of parameters:** In some of the tables there is no number of parameters (Table 3), and in others tables the proposed method has a higher amount of parameters (Table 12), this is even more important for Table 2. At least part of the difference between the first two rows (Dense kernel vs Multiresolution) can be explain by additional parameters. Can the authors conduct the experiments in Table 2 where the number of parameters is normalized across rows?

3. Measuring the **impact of hyperparameters and ablation studies**: Conducting additional ablation studies could provide more insights into the robustness and design principles of the method. Examples include ablating the number of resolutions, the initial resolution, the decay factor, and conducting experiments with LayerNorm instead of BatchNorm, among others.
There is no theoretical justification provided, which would greatly enhance the paper. I suggest the authors explore the expressiveness, initialization, generalization, and inductive bias of MRconv in comparison to other convolutional layers such as SSMs, SGconv, and others.

4. **Insights on MRconv:**  It would be beneficial if the authors provided more insights gained during their research on MRconv. For instance, a subsection discussing what makes MRconv successful would be valuable. Potential reasons could include suitable inductive bias towards multi-resolution and better optimization properties arising from stable parameterization (e.g., Batch Normalization). Directly analyzing these factors could offer informative perspectives to the community and pave the way for further improvements.

5. **The Focus of the paper:** The authors focus on long sequence modeling, probably due to the results over LRA benchmark and the sub-quadratic dependencies in sequence length. However, I am not certain this is the strongest aspect of the method in practice. For instance, achieving SOTA results on the LRA benchmark could be due to a strong inductive bias toward locality and other properties, as explored by [5] and [6]. Perhaps the strongest aspect of the method is its ability to learn high and low frequencies through a stable and efficient parameterization, resulting in a very natural inductive bias.

6. **Extensions:** exploring extensions such as multi-axis variants (2D, 3D..), bidirectional design, gated convolutions, and the inner block design could enhance the paper.

7. **Novelty:** The method is relatively simple and can be described with just a few equations (as described in Algorithms 1,2,3). Although there are no particularly surprising ideas, the design choices are sound, the empirical evaluations are thorough, and the performance is nearly SOTA across several modalities. Therefore, I do not consider this limitation to be significant.

I am willing to raise my score if the concerns specified in this section will be sufficiently addressed during the rebuttal period.

[1] Simplifying and Understanding State Space Models with Diagonal Linear RNNs. Gupta et al.

[2] Mamba: Linear-Time Sequence Modeling with Selective State Spaces. Gu et al.

[3] Hungry Hungry Hippos: Towards Language Modeling with State Space Models. Fu et al.

[4] Mechanistic Design and Scaling of Hybrid Architectures. Poli et al.

[5] Viewing Transformers Through the Lens of Long Convolutions Layers. Zimerman et al.

[6] Never Train from Scratch: FAIR COMPARISON OF LONGSEQUENCE MODELS REQUIRES DATA-DRIVEN PRIORS. Amos et al.

**Questions:**

Please review the weaknesses, particularly those highlighted in W.1, W.2, W.4, and W.5.

**Limitations:**

The limitations section can be improved. Please refer to W.1.b for details on the failure cases of the layer.

---

> ### Author Rebuttal · Authors · 2024-08-06
>
> We appreciate the reviewer's useful feedback and address their weaknesses and questions below. We hope this resolves any of the outstanding concerns.
> ## 1. Results on NLP
> Although NLP is an important area of study, we have yet to explore the application of MRConv in its current form to language modelling. Previous purely convolutional methods, such as S4, have struggled to outperform attention-based architectures without additional input dependencies, such as gating used in Hyena and H3 or attention in MEGA. Additionally, [1] has theoretically proven that non-input-dependent gated convolutions cannot solve multi-query associate recall tasks without the hidden state dimension scaling linearly with the sequence length. As a result, we view the paper's **contributions in training and structuring long convolution kernels** as a promising initial step toward language modelling. We believe that **designing input-dependent architectures** that leverage multi-resolution convolutions for NLP presents a promising future research direction.
> ## 2. Efficiency Benchmarking
> When comparing FLOPs between different models, it's crucial to consider all computations to compare models with different architectures, such as the Swin transformer, which uses patching and self-attention rather than convolutions. Isolating only the convolution layer would render such comparisons challenging. Furthermore, when comparing FLOPs between ConvNeXt and MRConvNeXt, given both use an identical ConvNeXt backbone, the only discrepancy in FLOPs arises from altering the convolutional layers. Hence, we think it is intuitive how our 1D convolutions reduce the number of FLOPs compared to 2D convolutions.
> ## 3. Runtime Comparison
> Please see our Author's rebuttal at the top of the page and the attached PDF. Our results show that **MRConv is substantially faster than FlashAttention**. We note that MRConv, SGConv, Conv1D are all equivalent to a global convolution and hence runtimes are identical once the kernel has been computed and cached. Table 4 in our paper presents further runtime comparison, for which **MRConv is 1.5 and 1.3 times faster than S4 on ListOps and Image tasks**.
> ## 4. Normalized parameter counts
> Throughout our evaluation, we deliberately choose to focus on equating computational complexity between different models rather than parameter counts. For example, in Table 2, although each design ablation has a different number of parameters, each model maintains **identical computational complexity**, corresponding to a model with a fixed width and fixed number of global convolution layers. We find this to be a more natural point of comparison, which is more correlated with practical performance.
> ## 5. Additional ablation studies
> We want to thank the reviewer for their valuable suggestions. We have followed the reviewer's guidance and conducted the additional ablations which we will include in our camera-ready paper.
> ### 5.1 Initial resolution
>  We find that the initial resolution is an important hyperparameter that determines the number of resolutions but also performance on different data modalities. As a result, we provide additional ablation studies on the *ListOps* and *Image* LRA tasks where we vary the initial kernel size.
> > |Initial Kernel Size|Num Resolutions|Accuracy|
> |-----|-----|-----|
> *Listops - Fourier Kernel*
> |1|11|61.80|
> |2|10|62.40|
> |4|9|61.10|
> |8|8|60.95|
> *Cifar - Fourier Kernel*
> |8|8|86.69|
> |16|7|88.39|
> |32|6|88.55|
> ### 5.2 Decay factor
> A key factor of our work is that we don't use a fixed decay factor but that we learn one implicitly by **learning the weighted sum** of multi-resolution kernels. In Table 2, we show that learning the weighted sum of kernels improved performance over dense kernels by 3.9\% and 3.75\% on the *ListOps* and *Image* tasks respectively.
> ### 5.3 LayerNorm
> **LayerNorm is not amenable to structural reparameterization** as it remains a nonlinear operation at inference and hence we don't consider it to normalize each multi-resolution convolution.
> ### 5.4 Initialization
> A key advantage of MRConv is its **simple initialization of convolution kernels**, unlike S4 and S4D which require intricate HiPPO theory to initialize them correctly.
> ## 6. Insights on MRConv
> In Section 5.1 'Resolution Analysis' we analyse the inductive bias of MRConv to learn different frequency kernels at different layers in the network. This is an inductive bias driven by our multi-resolution framework and ability to implicitly learn the decay rate via the weighted sum of kernels. Further in Section 5.1 'MRConv Design' we perform a detailed ablation study showing how each component of our multiresolution framework improves performance. In particular, we highlight the importance of BatchNorm and how normalizing the activations from each convolution before summing is imperative to performance. We also include additional convergence plots to accompany these ablations in our author rebuttal which further emphasize how our design features enable faster convergence and better generalisation by preventing overfitting. We will make sure to include a summary of our insights in the revised paper.
> ## 7. Focus of the paper
> We thank the reviewer for raising this point and clarify our motivation. The motivation for our work is to **parameterize global convolution kernels such that they have the correct inductive biases for long sequence modelling**. Indeed, the biggest problem with using convolutions for long sequence modelling is overfitting and we show that our parameterization not only **provides stable training** (see additional convergence plots in author rebuttal) but also **learns to prioritize local information** (Figure 3 in paper) equipping our kernels with the right inductive bias to make them **highly effective on long sequence modelling tasks**. We will make sure to highlight this fully in our revised paper!
>
> [1] Arora, Simran, et al. "Zoology: Measuring and improving recall in efficient language models." 2023

---

> > ### Comment · Reviewer_JNPq · 2024-08-12
> > **Official Comment by Reviewer JNPq**
> >
> > Thank you for the response.
> >
> > Some of my concerns have been addressed, particularly the ‘Runtime Comparison’ and the ‘Normalized Parameter Counts,’ which are essential details that make the ablation in Table 2 and the efficacy analysis much clearer. Please include these details in the final version of the paper.
> >
> > Additionally, I find the ‘Additional Ablation Studies’ and ‘Insights on MRConv’ in the above response to be informative, as well as other responses to the reviews (2.2 Diversify Optimization - pbnE, Implicit Kernel Parameterizations - wDz6), which are important additional ablation studies/comparisons.
> >
> > Given that some of my concerns have been addressed, I am raising my score and confidence to 5.
> >
> > ---
> >
> > **There are still several concerns and requests for clarification:**
> >
> > **Runtime Plots (PDF):**
> >
> > Could you add a curve of simple SSM with a similar model size to this figure? It could provide important details about the differences between sub-quadratic layers.
> >
> > **FLOPS comparison:**
> >
> > The paper compares FLOPs using the ConvNeXt backbone, but this is the only FLOPs comparison it provides. I believe this comparison can be misleading and may lead to **incorrect** conclusions when evaluating the FLOPs required by MRConv versus those needed by Attention, Conv2D, SGConv, and others.
> >
> > To clarify, imagine that 99.99% of the FLOPs in ConvNeXt come from the MLP (or Conv1x1), with only 0.01% from Conv2D/MRConv. Even if MRConv requires 10 times more FLOPs than Conv2D, the overall difference in FLOPs between the final models would be just about 0.1%. This small difference could **obscure** the true computational demands of MRConv.
> >
> >
> > **Results on NLP:**
> >
> > I understand that MRConv might struggle to outperform attention-based architectures. In fact, MRConv will likely struggle and perform quite poorly compared to attention-based methods, and that’s okay. Nevertheless, negative results can be valuable and informative, especially when compared to other variants (see Table 1 in the HGRN[1] paper as an example). I’m not asking you to provide SOTA, rather, I would like to see experiments that help us understand the role of multi-resolution inductive bias in NLP and identify any unique challenges, such as optimization issues or similar problems.
> >
> > > 5.3 LayerNorm: LayerNorm is not amenable to structural reparameterization as it remains a nonlinear operation at inference and hence we don't consider it to normalize each multi-resolution convolution.
> >
> > I understand that LayerNorm is much less efficient during inference. Nevertheless, I believe this ablation can shed more light on the performance and training dynamics of MRConv. In particular, BatchNorm is considered less stable than LayerNorm in some regimes and can introduce train-test discrepancy. Therefore, conducting ablations with LayerNorm could be informative and expose limitations of the proposed method (which is good, as it defines a clear direction for improvements).
> >
> > ---
> >
> >  If you plan to conduct some of the proposed experiments but may not be able to complete them by the end of the discussion period due to time constraints, please let me know.
> >
> > [1] Hierarchically Gated Recurrent Neural Network for Sequence Modeling.

---

> > > ### Author Response · Authors · 2024-08-13
> > >
> > > We thank the reviewer for their response and for raising their score. We answer their outstanding concerns and suggestions below.
> > > ## Runtime Plots
> > > We agree with reviewers suggestion and compute run time results for S4D from the s4 Github repository. Our results show that the computation time for S4D and MRConv are near identical as both utilize FlashFFTConvolutions, but **MRConv is faster due to S4D being slower to compute the convolution kernel** via Vandermonde matrix multiplications as opposed to simpler FFTs in MRConv. All timings below are given in ms. We will make sure to add the S4D runtime performance to our throughput figure and include this figure in our updated paper.
> > > |Model|L=1024|L=2048|L=4096|L=8192|L=16384|L=32768|
> > > |---|---|---|---|---|---|---|
> > > |MRConv|2.30|2.93|3.64|7.25|30.3|63.0|
> > > |S4D|2.45|4.01|6.12|12.1|39.7|84.3|
> > >
> > > ## FLOPs Comparison
> > > To provide extra clarity on the computational demands of MRConv, we provide a further **theoretical analysis on the number of FLOPs** required to compute a 1D FFT convolution and a regular 2D convolution. In all comparisons, we will use a batch size of 1 and a channel dimension of 1. An FFT convolutions involves: i) a forward FFT of the input, ii) an elementwise product between the input and the kernel, and iii) an inverse FFT of the resulting convolution. The total time complexity is $\mathcal{O}(2L\log L + L)$. In practice, we use the real-valued FFT implemented via the Cooley-Tukey algorithm, which uses $2L \log L$ FLOPs [1]. On the other hand, regular 2D depthwise separable convolutions with a kernel size of $k$ have a complexity of $\mathcal{O}(k^2 H W)$, requiring $2 k^2 H W$ FLOPs with 2 FLOPs needed for each multiply-accumulate operation. Next, we calculate the number of FLOPs to compute a 2D 7x7 convolution, as utilized in ConvNeXt and a global 1D convolution as utilized in MRConv, for a sequence of increasingly larger (flattened) images. Our results show that our **1D convolutions use fewer FLOPs than the comparable 2D convolution used by ConvNeXt**.
> > > |Conv Type|Conv Size|(16,16)/(256,)|(32,32)/(1024,)|(64,64)/(4096,)|(128,128)/(16384,)|(256,256)/(65536,)|(512,512)/(262144,)|
> > > |---|---|---|---|---|---|---|---|
> > > |2D|$7\times 7$|25.1K|100K|401K|1.61M|6.42M|25.7M|
> > > |1D|$L$|8.7K|43.0K|205K|0.95M|4.32M|19.4M|
> > >
> > > We hope that this provides a clearer comparison between 1D FFT convolutions and 2D convolutions and we will make sure to add this to our paper. We thank the reviewer for emphasising this suggestion as we believe that this significantly improves the clarity of work!
> > >
> > > ## Results on NLP
> > > We agree with the reviewer and appreciate that negative results on language modelling using MRConv can be both valuable and informative. We too are particularly interested in the multi-resolution inductive bias on information dense data such as text and have performed **preliminary experiments** looking at the rate of kernel decay with depth in MRConv by plotting the $\alpha$ values corresponding to each multi-resolution kernel at each layer in the network (see Table 3b). Regarding additional experiments on language modelling, we won't be able to complete these by the end of the author-reviewer rebuttal period. Looking at the experiments conducted in HGRN, they required use of 8 x 80GB A100 GPUs. We perform almost all of our experiments on a single 40GB A100. Acquiring the compute to perform such experiments is likely to be costly and take some time. We firmly believe though that applying MRConv to language data is an **exciting future direction**, in terms of performance and as a means of uncovering learning biases and optimization challenges in language modelling and it is a path we are dedicated to following in future work.
> > >
> > > ## LayerNorm Ablation
> > > We provide a further ablation where we substitute out BatchNorm for LayerNorm. We find that **performance of LayerNorm comparable, if not slightly worse, than BatchNorm** without hyperparameter tuning. This further emphasises the **suitability of BatchNorm for structural reparameterization**, on top of being able to reparameterize BatchNorm at inference.
> > > *ListOps - Fourier Kernel*
> > > |Norm Type|Accuracy|
> > > |---|---|
> > > |BatchNorm|62.40|
> > > |LayerNorm|60.10|
> > >
> > >
> > > *Image - Fourier Kernel*
> > > |Norm Type|Accuracy|
> > > |---|---|
> > > |BatchNorm|89.30|
> > > |LayerNOrm|87.80|
> > >
> > > We wish to thank the reviewer for their suggestions which have helped improve our manuscript enormously. We hope these adjustments and explanations adequately address the issues highlighted and kindly enquire if our replies and additional experiments have led them to change the paper’s score. If there are any remaining questions, we are fully prepared to address them accordingly at the last minute. We eagerly looking for your additional feedback on our response!
> > >
> > > [1] Arunachalam, S., Khairnar, S.M. and Desale, B.S., 2013. The fast Fourier transform algorithm and its application in digital image processing. New J Chem

---

> > > > ### Comment · Reviewer_JNPq · 2024-08-14
> > > >
> > > > Thank you for your response. I appreciate the authors' new analysis (LayerNorm Ablation, FLOPs Comparison and Runtime Plots) and their efforts to address my concerns, which have made my decision much easier. I also believe that, beyond addressing the reviewers' concerns, the response and discussion (with all the reviewers) have improved the paper, making it clearer and more informative. Consequently, I have increased the score from 5 to 6.
> > > >
> > > > Regarding the NLP experiments, training such models on Wikitext should take less than a day on a single A100 GPU (rather than eight), assuming the context length is not too long (512, for example). However, while this point is interesting, it is not crucial given the extensive ablations already presented in the paper and the discussion. I mention it only for clarity.
> > > >
> > > > Best wishes

---

### Official Review · Reviewer_pbnE · 2024-07-13

**Soundness:** 4
**Presentation:** 3
**Contribution:** 3
**Rating:** 6
**Confidence:** 4

**Summary:**

This paper proposes MRConv, a new way to parameterize global convolutional kernels for long sequence modeling. MRConv pads all sub-kernels to the same length and aggregate outputs of sub-kernels with batchnorm and linear rescaling. Three different kernel initializations are explored. Experiments on Long Range Arena and speech and image classification tasks show that MRConv is more computationally efficient than SSM baselines and slightly better than SGConv in accuracy.

**Strengths:**

1. MRConv achieves state-of-the-art performance on LRA benchmark and several classification tasks compared to prior long sequence modeling methods.
2. MRConv maintains inference speed advantage over SSM baselines with added modeling complexity compared to SGConv.
3. Ablations show that Fourier kernels are the most robust across different tasks, saving users' time to choose kernels themselves.

**Weaknesses:**

1. Lack experiments on generative tasks like language modeling, speech and image synthesis. These are more challenging tasks where many SSMs like Mamba [1] are already tested on.
2. Multi resolutional convolution methods including SGConv and MRConv lack theoretical analysis on its expressivity. Is it more powerful than SSM or are they equivalent? A clarification is needed.
3. Table 8 shows that MRConv needs additional modification to work on Path-X task. Task-specific modification on the model diminishes its general applicability to a wider range of tasks.

[1] Gu, Albert, and Tri Dao. "Mamba: Linear-time sequence modeling with selective state spaces." arXiv preprint arXiv:2312.00752 (2023).

**Questions:**

1. Line 107-110 says BatchNorm is added since output statistics are different for different kernel size. But BatchNorm is normalizing over instances in the same batch, the difference of kernel size still remains. Am I understanding this correctly or is there other reason behind BatchNorm?
2. Line 159, what is this $k$ ? The number of parameters in the kernel?
3. Is it possible to have all three kernel parameterization together using gating mechanism to choose which kernel to use?

**Limitations:**

The authors have adequately addressed the limitations.

---

> ### Author Rebuttal · Authors · 2024-08-06
>
> We thank the reviewer for their support of our paper and their insightful feedback which leaves plenty of room for future work. Below we answer the questions raised by the reviewer.
> ## 1. Comparison with SSMs
> As mentioned in our introduction, SSMs such as S4 and S4D can be represented equivalently as a global convolution, with the kernel implicitly parameterized by the SSM system matrices (refer to Equation 3). Hence, in theory, SSMs and global convolution methods like MRConv perform the same operation. Therefore, **differences in expressivity depend on how different methods implicitly parameterize the convolution kernel**: SSMs do so via system matrices, while MRConv does so through the weighted combination of multi-resolution kernels. Under certain parameterizations, we establish a **direct theoretical equivalence between SSMs and Fourier kernels**, where the SSM parameters define a kernel as the weighted sum of truncated Fourier basis functions (see Appendix B.3). However, it is not immediately clear how to reparameterize multi-resolution SSMs when computed in recurrent form. Generally, measuring the expressivity of different convolution kernels is a non-trivial task. Consequently, we **evaluate expressivity based on empirical results** from benchmark tasks, and our findings indicate that **MRConv outperforms S4 and SGConv on LRA, sCIFAR, and Speech Commands.**
> ## 2. BatchNorm
> We incorporate BatchNorm primarily for 2 reasons
> ### 2.1 Preserve Variance
> Firstly we use BatchNorm to ensure that the **variance of the output from each convolution remains consistent** regardless of the kernel length. We find this is crucial for learning the weighted sum of kernels. Indeed, the addition of BatchNorm significantly **enhances performance**, as evidenced in Table 2 of the paper, and **accelerates convergence**, as illustrated in our new convergence plots in our one-page attachment in the author's rebuttal. We note that other types of normalization, such as LayerNorm, remain non-linear during inference and therefore are not amenable to structural reparameterization unlike BatchNorm.
> ### 2.2 Diversify Optimization
> Further, BatchNorm is also used to **diversify optimization by introducing training-time non-linearity** by dividing the output of each convolution by its standard deviation [1]. Consequently, during backpropagation, we compute gradients which cannot be calculated by differentiating a global kernel that has already undergone reparameterization. We provide an extra ablation study where we replace BatchNorm with normalization by a constant factor, computed as the norm of the kernel over the sequence length at initialization. Our results highlight that while normalization by a constant factor improves performance compared to no normalization, it still falls short of the benefits of using BatchNorm.
>
> > *ListOps - Fourier Kernel*
> |Norm Type|Accuracy|
> |-----|-----|
> |BatchNorm|62.40|
> |$1/\|k\|$|61.05|
>
> > *Image - Fourier Kernel*
> |Norm Type|Accuracy|
> |-----|-----|
> |BatchNorm|89.30|
> |$1/\|k\|$|87.72|
>
> ## 3. Typo
> We would like to thank the reviewer for bringing this to our attention. In this context, the parameter $k$ denotes the number of non-zero elements in the dilated convolution kernel. In the final version, we will rectify the error and clarify the text. Thank you once again for identifying this oversight!
> ## 4. Reparameterization via Gating
> This suggestion is excellent. Firstly, we found that linear rescaling works well when reparameterizing multiple kernels of the same length (see Section 3.1), eliminating the need for BatchNorm. This allows us to reparameterize many different kernels with differing parameterizations during training at no extra cost. Our initial investigations combining both Fourier and Sparse kernels suggest that this is a highly effective parameterization, delivering competitive results on LRA and ImageNet classification. Secondly, while we currently learn a fixed set of weights to linearly combine each kernel, gating could be a highly effective method for combining kernels in an input-dependent manner. We believe this could effectively introduce input dependency into our model, potentially aiding the application of MRConv to more complex data modalities, such as text. We leave this suggestion for future work but thank the reviewer for their inspiring comment!
>
> [1] Ding, Xiaohan, et al. "Repvgg: Making vgg-style convnets great again." 2021.

---

### Official Review · Reviewer_wDz6 · 2024-07-15

**Soundness:** 3
**Presentation:** 3
**Contribution:** 3
**Rating:** 7
**Confidence:** 4

**Summary:**

This paper introduces reparameterized multi-resolution convolutions, a multi-resolution approach for the parameterization of global convolutional kernels for long sequence modeling. Their idea is to view long convolutional kernels as the combination of kernels at multiple resolutions, each with the same number of parameters.

The authors evaluate their proposed MRConv on multiple long sequence modeling tasks such as LRA. In addition, they show  interesting performance results on ImageNet classification, when replacing 2D convolutions with 1D MRConv layers.

**Strengths:**

- The authors propose a novel way to parameterize convolutional kernels in a multi-resolution fashion.
- The authors demonstrate that their parameterization leads to interesting performance gains across multiple long-term dependency tasks.

**Weaknesses:**

- It is clear that the main weakness of the method is the additional memory and time costs during training. However, very little is mentioned about this. I believe this is very important, as the costs will scale with the number of sub kernels considered, which in turn, as far as I understand, are proportional to the length of the input itself. I think that making this clear in the paper is of utmost importance, as this is likely the main factor that would prevent this method to be adopted in practice. Both a complexity analysis and throughput analyses should be added. Due to the dependency between the number of subkernels and the input sequence length, I am afraid that the proposed model might scale quadratically during training wrt sequence length.

- The authors make multiple statements that lack references from which I am not entirely sure they are entirely correct. I would appreciate it if the authors could support these claims better. For example:
  - ln 19. “... due to training instabilities, inadequate inductive biases to prioritize important contextual information and prohibitive computational complexities.”
  - Ln 35. “... a decaying kernel structure, a classical design principle from signal processing, s.t. weights closer to the input are larger than ones further away.” -> I feel this might be a bit of a stretch. Please add references or clarify where this is coming from.
  - Ln 81. “Alternative implicit kernels can also be designed so that they can be computed in O(L) time making them faster than SSMs implemented as a global conv.” -> Please add references.

- The notation in the paper should be made consistent. Both bold and non-bold symbols are both used to define matrices, vectors and scalars, e.g., $\alpha$, $\boldsymbol{\alpha}$.

**Questions:**

- There are no ablations wrt the decay filters in Table 2.

- There are multiple decisions taken in the paper that are not entirely clear to me. I would like to understand why these decisions are taken, given that these, in my opinion, overcomplicate the method and hinder its scalability.
  - First, it is unclear to me why the BN is required after the convolution of each kernel. In principle, if the purpose of BN is to normalize the kernel, then it is simpler to normalize the convolutional kernels directly. By doing so, then all kernels can be combined pre-convolution, and one single convolutional operation would be needed. On the other hand, if the purpose of BN is to make sure the variance of the input and the output is preserved, then there’s also an easier solution: the kernel can be multiplied by a constant coefficient proportional to the inverse of the fan_in of the kernel, as proposed in [1, 2].
  - Note that if the previous suggestions are followed, then the whole convolutional layer would be computed in O(N log N) time both during training and inference. Also, it would be really simple to combine kernels on the Fourier domain, which in my opinion would also make the method much more appealing. In fact, only the input would need to go through the forward FFT operation.
  - Also, implicit kernel parameterizations should be more expressive than dilated and sparse parameterizations. Is there a reason for not selecting this parameterization?

- There is something I do not understand about the way the kernels are merged. You state that you “construct global kernels as the learnable summation of normalized multi-resolution kernels” - Ln 85. But if I look at the definition, it seems that the $alpha$ values refer rather to a learnable exponential decay of the kernels. When does the learnable summation take place? Does $alpha$ obey both purposes –that of exp. decreasing the kernel and reweighting it? Or are the learnable parameters of the BN layer the ones responsible for this?

[1] https://arxiv.org/abs/2301.10540

[2] https://arxiv.org/abs/2312.08399

**Limitations:**

The authors clearly state a limitations section for the method. However, as mentioned in the weaknesses section, the paper lacks descriptions regarding how big these limitations are in practice.

### Conclusion

Whilst I very much like the idea of MR-parameterized convolutional kernels, and acknowledge the novelty of the paper, I am concerned about the scalability -and therefore the impact- of the proposed method. Therefore, I am hesitant to support acceptance. However, I want to note explicitly that I think that this paper could be very impactful, if proper adjustments are made. I am happy to increase my score should my concerns and comments be addressed.

---

> ### Author Rebuttal · Authors · 2024-08-06
>
> Thank you for your overall supportive review of our work. The reviewer asked several insightful and detailed questions about our proposed method, which we will respond to in order.
> ## 1. Complexity analysis
> We agree with the reviewer that, compared to inference complexity, our training complexity is more costly. However, we do not see this as a limitation of our method because:
> i) it is **more important to have lower inference complexity** than training complexity in practical applications and
> ii) our **training complexity is still considerably more efficient than self-attention** in theory. In light of the reviewer's comments, we show that a complexity analysis highlights the theoretical benefits of our model regarding training and inference complexity. Assuming we have $\log_2(L/l_0)$ resolutions, the computational complexity during training to compute the convolutions is $\mathcal{O}(L \log_2(L/l_0) \log_2(L))$. In the most computationally demanding setting where the initial resolution $l_0=1$ the computational cost during training is $\mathcal{O}(L \log_2^2(L))$, which is **subquadratic wrt. sequence length**. Hence, even when training individual kernels in parallel, our model is **theoretically more efficient than self-attention mechanisms**, which scale quadratically with sequence length during training. In the camera-ready version, we will add this complexity analysis and a more detailed discussion of the computational costs incurred during training.
> ## 2. Runtime Comparison
> Please see our Author's rebuttal at the top of the page and the attached pdf. Our results show that **even our non-reparameterized kernels are efficient and remain computationally faster than FlashAttention**, aligning with our theoretical complexity calculations
> ## 3. Decay Filters
> Ablations on LRA wrt. decay filters can be found in Table 1.
> ## 4. Use of BatchNorm
> BatchNorm serves multiple purposes in our parameterization.
> ### 4.1 Preserve Variance
> Firstly, it ensures that the variance of the output of each convolution equals that of the input, **improving performance**, as evidenced in our design ablations in Table 2 and **accelerating convergence**, as illustrated in our new convergence plots in our one-page attachment in the author's rebuttal
> ### 4.2 Diversify Optimization
> Secondly, BatchNorm **diversifies optimization dynamics** by introducing additional training-time non-linearity, leading to gradients that cannot be replicated by an equivalently reparameterized kernel [1]. In response to the reviewer's suggestion, we conducted an additional ablation study, substituting BatchNorm with **normalization by a constant factor**, computed as the norm of the kernel over the sequence length at initialization. Our results indicate that while normalization by a constant factor improves performance compared to no normalization, it still **falls short of the benefits of using BatchNorm**. Nonetheless, this presents a promising research avenue, and we appreciate the reviewer for their valuable input!
>
> > *ListOps - Fourier Kernel*
> |Norm Type|Accuracy|
> |-----|-----|
> |BatchNorm|62.40|
> |$1/\|k\|$|61.05|
>
> > *Image - Fourier Kernel*
> |Norm Type|Accuracy|
> |-----|-----|
> |BatchNorm|89.30|
> |$1/\|k\|$|87.72|
>
> ## 5. Combining kernels in the Fourier space
> Combining the kernels in the Fourier space to avoid excessive FFTs is an attractive proposition. However, a key challenge arises from the fact that each multiresolution kernel has a different length, which means that each kernel is defined by an inverse FFT of different length. As a result, it's not straightforward to parameterize all kernels in a single Fourier basis defined over the longest kernel length. We also note, that even in our current parameterization **we only require a single FFT of the input**, as it is reused in each multi-resolution convolution.
> ## 6. Implicit kernel parameterizations
> This is a very interesting question and boils down to what is meant by expressivity of parameterization. In our work we base expressivity on 2 factors: i) the **number of parameters** used to define the kernel; the number of Fourier modes or the number of non-zero values in the sparse parameterizations and ii) the **kernels inductive bias**; Fourier kernels are very smooth whereas dilated kernels are very rough. We choose to focus on the kernels inductive bias and show that different kernels perform better on different data modalities (see Section 5.1 'Kernel parameterization').  As suggested we ran an extra ablation study where we parameterize the kernel as a small MLP as used in CKConv and Hyena. Our results show that the **inductive biases of the Fourier and dilated kernels are better suited to the *ListOps* and *Image* tasks** in the LRA benchmark. It would be very interesting to consider different implicit parameterizations in future work.
> > *ListOps*
> |Kernel Type|Params|Accuracy|
> |---|---|---|
> |Dilated|759K|59.25|
> |Fourier|332K|62.40|
> |Fourier+Sparse|420K|62.25|
> |MLP|580K|60.08|
> > *Image*
> |Kernel Type|Params|Accuracy|
> |---|---|---|
> |Dilated|3.5 M|90.37|
> |Fourier|3.8M|88.55|
> |Fourier+Sparse|4.0M|89.07|
> |MLP|3.6 M|85.71|
>
> ## 7. Learnable Summation
> The parameter $\alpha$ represents the weight assigned to each convolution kernel before summation. Upon reviewing equations 12 and 13 in the paper, it appears that the reviewer may have misconstrued $k_i$ as the first element of some kernel $k$, when in fact $k_i$ corresponds to the convolution kernel of length $l_i$ at resolution $i. Consequently, **$\alpha_i$ does indeed influence the decay rate** of the reparameterized kernel. Generally, **larger values of $\alpha$ are associated with shorter kernels**, while smaller values are associated with longer ones, as demonstrated in our ablation study in Figure 3. This effect is evident in Figure 5, where we visualize the kernels learned in our ImageNet experiments.
>
> [1] Ding, Xiaohan, et al. "Repvgg: Making vgg-style convnets great again." 2021.

---

> ### Comment · Reviewer_wDz6 · 2024-08-12
>
> Dear authors,
>
> Thank you very much for your reply and additional experiments. I am looking forward to your future work. Hopefully these suggestions will help in creating a faster MRConv version in the future.
>
> I understand your point about the method being faster than Transformers. But I'd still argue that the method can become, for example, dramatically slower than 1 resolution methods, specially during training. I do not think that this "deficiency" is a deal breaker in terms of the value and possible impact of the paper. However, I do think that making this **very** clear is important.
>
> Under the promise that the authors will make the limitations of the paper clear in the final version of the paper, I am happy to support acceptance. Under this promise, I have now increased my score to 7.

---

### Official Review · Reviewer_8iUW · 2024-07-18

**Soundness:** 3
**Presentation:** 2
**Contribution:** 3
**Rating:** 5
**Confidence:** 5

**Summary:**

This paper tries to tackle the training challenge of long convolutions with reparameterized multi-resolution convolutions (MRConv), which parameterizes global convolutional kernels for long-sequence modeling. The authors introduce learnable kernel decay to learn expressive long-range kernels that perform well across various data modalities. Extensive experiments on the Long Range Arena verify the effectiveness of the proposed MRConv across various tasks in comparison to state-space-based and attention-based models. Moreover, the proposed 1D MRConv can replace 2D convolutions for ImageNet classification tasks and yield better performances.

**Strengths:**

(S1) This paper improves the training challenge of SSMs with an intuitive and efficient design. The proposed three types of low-rank kernel parameterization methods are suitable and effective for long-sequence modeling scenarios.

(S2) Compared to popular long-sequence modeling works, the proposed MRConv can achive state-of-the-art performances on some LRA benchmarks. Additional experiments on ImageNet-1K also verify that MRConv can be a general design and further benefit vision communities to some extent.

(S3) The overall presentation is easy to follow, and the structure is clear and easy to read.

**Weaknesses:**

(W1) The proposed MRConv is not novel compared to. The idea and techniques of structural reparameterization of convolution kernels are widely used in modern vision architectures (like RepLKNet variants) and have been recently adopted to long-sequence scenarios in CHELA [1]. As for the three proposed low-rank kernel parameterization strategies, the dilated kernels and sparse kernels have been adopted in vision networks VAN variants [2, 3] and SlaK variants [4, 5], and the Fourier kernels are somewhat novel (but intuitive according to S4 backgrounds).

(W2) The authors should propose a final practical version among three designed versions of SGConv/MRConv, considering the performances, parameters & FLOPs, and generalization across tasks. It can be comprehensive to readers by providing all results (including ablations) of three versions in all comparison tables. However, I am confused about choosing a general and efficient implementation based on these results because I cannot find a version that consistently outperforms others. Therefore, I suggest the authors provide some summed conclusions (like take-home messages) and a final version.

(W3) Drawbacks in experiments. Firstly, some recently proposed works like [1, 6] are missing in the comparison experiments, which weakens the comprehensive evaluations. Secondly, MRConv variants can only achieve competitive results as previous works on LRA benchmarks, and the performance gains are not significant. Moreover, this work is motivated to improve the training instabilities, and some verifications on this aspect are missing (e.g., convergence speeds).

### Reference

[1] Zicheng Liu, et al. "Short-Long Convolutions Help Hardware-Efficient Linear Attention to Focus on Long Sequences." ICML, 2024.

[2] Menghao Guo, et al. “Visual Attention Network.” CVMJ, 2023.

[3] Siyuan Li, et al. “MogaNet: Efficient Multi-order Gated Aggregation Network.” ICLR, 2024.

[4] Shiwei Liu, et al. “More convnets in the 2020s: Scaling up kernels beyond 51x51 using sparsity.” ICLR, 2023.

[5] Honghao Chen, et al. “PeLK: Parameter-efficient Large Kernel ConvNets with Peripheral Convolution.” CVPR, 2024.

[6] Songlin Yang, et al. “Gated Linear Attention Transformers with Hardware-Efficient Training.” ICML, 2024.

================== Post-rebuttal Feedback ==================

Considering the authors' rebuttal and other reviews during the rebuttal period, my concerns have been well addressed and I believe this work meets the standard of NeurIPS. I suggest the authors further polish the manuscript with these valuable explanations and additional results in the main text or the appendix to make it an interesting and solid work.

**Questions:**

(Q1) Are there any rules for selecting the multi-resolution kernels across different tasks? As detailed in D.2.1, the hyper-parameters of multiple kernels sweep for all experiments, and it can be important for the practical usage of MRConv. As shown in D.5.2, the learned kernels on ImageNet-1K are visualized in the Fourier domain, and I suggest the authors conduct a similar analysis with more tasks, which might provide some interesting findings.

(Q2) Are there any training tricks for long convolution kernels in addition to the proposed MRConv? I found the authors utilize the different learning rates for long kernels and other parameters. The authors can provide more ablations of training stabilities (or convergence speeds).

(Q3) As the proposed MRConv requires BatchNorm, does it conflict with some LayerNorm when plugging MRConv into existing architectures (e.g., ConvNeXt variants)?

**Limitations:**

Some limitations have been considered in the conclusions section, but more issues of practical usage can be discussed based on the weaknesses I mentioned.

Overall, despite this manuscript having some merits, I appreciate that the weaknesses and questions I mentioned prevent it from accepting at the current stage. I am looking forwards to the authors’ feedback. I will consider raising my score if all these concerns are well addressed or perswading me.

---

> ### Author Rebuttal · Authors · 2024-08-06
>
> We thank the reviewer for pointing out several points of inclarity in the existing manuscript, which we seek to address below:
> ## 1. Novelty of MRConv
> Whilst structural reparameterization has been used in computer vision, we believe that its application in long-sequence modelling has not yet been effectively demonstrated. Therefore, our primary motivation is to adapt structural reparameterization for effective long-sequence modelling. Specifically, we want to highlight key novelties between MRConv and  previous methods.
> ### 1.1 Linear Reparameterization
> CHELA places a short and long convolution in **sequentially**, reparameterizing them  into a single kernel by **convolving** the kernels together. On the other hand, we compute convolutions in **parallel** and then reparameterize them to a single kernel by **summing** the kernels together. Further, CHELA incorporates a **non-linear activation** function between its short and long convolutions, which cannot be merged into one convolution. In contrast, MRConv can be properly reparameterized into convolution due to its **linearity**.
> ### 1.2 Weighted Combination
> Prior use of structural reparameterization in 2D vision tasks has simply summed the kernels after normalization, however for 1D long sequence modelling prioritization of local information is an important inductive bias to prevent overfitting and we are the first to propose **learning the linear combination of kernels via gradient descent**, the importance of which we highlight in Figure 3.
> ### 1.3 Multi-resolution Fourier parameterization
> We are the first to consider using a multi-resolution low-rank kernel parameterization in the **Fourier domain** for structural reparameterization. We note that our method is also different to S4 which considers global basis functions, whereas we sum kernels of increasing length but decreasing frequency.
> ## 2. Baseline comparisons
> We thank the reviewer for highlighting new work that also uses structural reparameterization. We note that some suggested works were not, or only just available online before the NeurIPS deadline, **including CHELA, which was uploaded to Arxiv after the NeurIPS deadline**, making comparison impossible in our original manuscript. Nevertheless, we will include the discussion in our camera-ready version. Further, the motivation of the ImageNet experiments is to show that long 1D convolutions can be a fast and effective alternative to 2D convolutions. We feel this message would be lost if we were to start comparing against new vision models which also make changes to the backbone ConvNeXt architecture and not just the convolution. For the final version, we will include these baseline comparisons to better represent the current state-of-the-art in ImageNet classification. We will also include a detailed discussion and comparison with CHELA, which wasn't possible to provide in our original submission.
> ## 3. Convergence Plots
> Following the reviewers' suggestions, we have provided convergence plots in our one-page attachment, which **highlight the improvements in training stability and convergence of MRConv**. In particular, we would like to thank the reviewer for this suggestion as the plots significantly improve the clarity of our ablations, highlighting MRConv's ability to avoid overfitting and strong generalisation and we will make sure to include them in the camera-ready version of our paper.
> ## 4. Kernel Selection.
> In Section 5.1, we conducted an ablation study to evaluate each kernel parameterization on different LRA tasks (refer to Table 1). Our findings indicate that the performance of kernel parameterizations varies depending on the data modality. For instance, we observed that "Dilated kernels perform exceptionally well on the Image task" (Line 203), while "on information-dense tasks such as ListOps, Fourier kernels perform better," as we hypothesize that "the sparse dilated pattern of dilated kernels is prone to skipping important tokens" (Line 204). These results suggest that the **performance of kernel parameterizations is influenced by the smoothness and modality of the training data**, indicating that **there is no one-size-fits-all solution**. This observation is consistent with previous results in [1] which show that different basis functions are optimal for different LRA tasks. When faced with uncertainty, we note that **'Fourier kernels perform the best on average'** (Line 206) and recommend using them as a strong starting baseline. In our revised paper, we plan to include a detailed discussion on kernel selection in different practical scenarios. We believe kernel selection is a promising future research direction that can further improve our framework and appreciate the reviewer for this suggestion!
> ## 5. Training Tricks
> We are happy the reviewer raised this question as the beauty of our method is in its simplicity in training. There are **no specific tricks required** to train MRConv. Using different learning rates in long sequence modelling is a common practice, widely employed in advanced architecture papers such as S4, S4D, S5, LongConv, and MEGA. Furthermore, compared to these architectures, we found that the beauty of MRConv lies in its ease of use; it is **less sensitive to hyperparameter tuning** and **does not require special initialization** such as HiPPO as used by S4.
> ## 6. Conflicts with BatchNorm and LayerNorm
> We find this not to be an issue. We want to highlight that although we use BatchNorm inside MRConv, the MRConv block **can be directly plugged into existing architectures that use LayerNorm**. For example, in our long-arena experiment in Table 1, the backbone architecture contains LayerNorms, and using MRConv still achieves the best performance, which demonstrates that MRConv can be easily integrated. We will further clarify this point in the camera-ready version.
>
> [1] Gu, Albert, et al. "How to train your hippo: State space models with generalized orthogonal basis projections." 2022.

---

> > ### Comment · Reviewer_8iUW · 2024-08-12
> > **Feedback to Authors' Rebuttal**
> >
> > Thanks for the detailed rebuttal for the authors. Some of my concerns were well tackled and there are further questions to those I think should be further explained.
> >
> > (W1) As for the novelty, the authors' rebuttal is reasonable to some extent. However, I suggest more comparison experiments or discussions with these methods to verify the priority of the designed methods. These existing works should not be overlooked despite the fact that most of them were originally proposed for 2D convolution networks.
> >
> > (W2) I cannot find any reply to W2. Could the authors provide more metrics that reflect computational efficiency, e.g., the number of parameters, FLOPS, and training times (is Figure 2 in the rebuttal PDF tackling this issue)? How to choose the final version among the three proposed implementations? Please tackle my concerns point-by-point!
> >
> > (W3) As for the comparison issues, I agreed with the authors' clarification. Some recently proposed methods like CHELA [1] and GLA [6] could be added to the appendix to make the comparison experiments more comprehensive. As for the convergence plots in the rebuttal PDF, the authors should explain the convergence curves of the three versions. For example, why does BatchNorm (MRConv) achieve inferior training accuracy while yielding better testing accuracy in Figure 1(a)?
> >
> > Overall, I am not quite satisfied with the authors' rebuttal. The authors should tackle all concerns in a well-arranged form. It is better to use the corresponding marks to indicate the question (e.g., W1, W2 to indicate weaknesses or summarize the question to prevent misunderstanding) and answer and use a new stating a new serial number to mark figures in the rebuttal PDF. If the authors do not make it easy for the reviewers to view the manuscript and rebuttal materials, it should not be expected that the reviewers spend a lot of time carefully understanding the rebuttal material when they have to review six or more papers. Therefore, I kept my score unchanged at the current stage.

---

> > > ### Author Response · Authors · 2024-08-13
> > >
> > > We thank the reviewer for their further suggestions on how to improve our paper. We structure our responses in the manner outlined by the reviewer to enhance readability, providing responses to W1, W2 and W3 in 3 separate comments below.
> > > # W1
> > > As the reviewer's suggestion, we update our discussion and experimental evidence on the benefits of our proposed use of i) linear reparameterization, ii) weighted combinations of multi-resolution kernels and iii) Fourier parameterized kernels in the context of existing work.
> > > ## i. Linear Reparameterization
> > > Firstly, we believe that our research provides compelling evidence of the success of reparameterizing kernels of different lengths through summation after training in parallel. In our Ablation study (Table 2), we demonstrate **a significant performance improvement of 13.5\% and 5.6\% when employing the sum of multi-resolution kernels** instead of dense kernels on the ListOps and Image LRA tasks. Additionally, by leveraging the linearity of BatchNorm at inference, we can reparameterize all our multi-resolution kernels into a single kernel, resulting in a substantial increase in throughput. Our findings show **throughput increases of $3.75\times$ and $2.17\times$ for the ListOps and Image LRA tasks**, respectively. These results are shown in Table 4b and further evidence of increased throughput can be found in our one-page pdf attachment.
> > > ## ii. Weighted Combination
> > > We are the first to explore the concept of learning a weighted summation of kernels of varying sizes, a novel approach in the field of vision where kernels are traditionally just summed without weighting. To assess the relevance of this feature in long-sequence modelling, we conducted an additional test where we applied our multi-resolution convolutions using a simple sum without learned weighting. The results revealed that **giving equal weight to each kernel failed to improve performance from initialization**, resulting in a significant drop. This underscores the importance of our weighted summation in learning an effective kernel decay, highlighting a **notable difference between reparameterizing in the vision and sequence modelling domains**. We will incorporate these findings into our Ablations in Table 2 and provide a detailed discussion in Section 5.1.
> > >
> > > *ListOps - Fourier Kernel*
> > > |Norm Type|Accuracy|
> > > |---|---|
> > > |Weighted Sum|62.40|
> > > |Sum|19.09|
> > >
> > > *Image - Fourier Kernel*
> > > |Norm Type|Accuracy|
> > > |---|---|
> > > |Weighted Sum|89.30|
> > > |Sum|17.84|
> > >
> > > ## iii. Multi-resolution Fourier parameterization
> > > We are the first to consider using a multi-resolution low-rank kernel parameterization in the Fourier domain for structural reparameterization. The **performance of our Fourier kernels is evaluated in our LRA Ablations in Table 1, achieving the highest average score of 87.84\%**, out-performing several powerful linear time-transformers with the same computational budget (see Table 4b in the paper for more results regarding comparison between Fourier MRConv and transformers utilizing linear attention mechanisms).
> > >
> > > We will update our paper accordingly to highlight the effectiveness of our 3 components introduced in MRConv on 1D sequence modelling tasks, in particular in relation to reparameterization schemes used in 2D computer vision. We also plan to **add disucssion in our related work on 2D structural reparamterization in computer vision**. We thank the reviewer for their suggestions.

---

> > > > ### Author Response · Authors · 2024-08-13
> > > >
> > > > # W2
> > > > Apologies if our original rebuttal was not clear enough for you. We organise our response to your concerns into separate points below.
> > > >
> > > > ## Parameters
> > > > All of our kernels use the **same number of parameters** at each resolution, and therefore the same number of parameters in total. We will add parameter counts corresponding to our LRA ablations of each kernel in our extended results table in the appendix in order to compare MRConv with other baseline models.
> > > >
> > > > ## FLOPs and Training Times
> > > > All of our proposed kernels have **near identical training and identical inference times**, as they are all implemented via FFT convolutions. The only exception is dilated kernels, whose training time throughput is marginally quicker due to fast dilated kernels, but once reparameterized we find it is quicker to implement via an FFT convolution due to the dilated pattern being no longer equally spaced. We will make sure to highlight this point for clarity in our paper.
> > > >
> > > > ## How to Choose Kernel
> > > > We restructure our response in Section 4 of our rebuttal 'Kernel Selection' and summarize the key points below:
> > > > 1. **Kernel Performance Varies by Data Modality**: In Section 5.1 we perform an ablation study evaluating each kernel parameterization on each LRA task (see Table 1), finding that different kernel parameterizations perform better or worse on different long-range arena tasks depending on the data modality. In particular *‘Dilated kernels perform exceptionally well on the Image task’* (Line 203), whilst *‘on information-dense tasks such as ListOps, Fourier kernels perform better'* where we hypothesise that *'the sparse dilated pattern of dilated kernels is prone to skipping important tokens’* (Line 204).
> > > > 2. **Influence of Data Smoothness and Modality:** We find it a significant result that different kernel parameterizations perform better on different tasks due to differences in the smoothness and modality of the training data and that there isn't necessarily a one-size-fits-all solution. This is further evidenced by how easily dense kernels, which can theoretically learn both smooth and sparse kernels, overfit to the training data and underperform MRConv when evaluated on the test data (see Table 2 in the paper and convergence plots in one-page pdf summary).
> > > > 3. **Fourier Kernels Perform Best on Average**: When faced with uncertainty, we note that *'Fourier kernels perform the best on average'* (Line 206) and recommended them as a reliable starting baseline when kernel choice is uncertain. Based off our results on LRA, sCIFAR and Speech Commands, we otherwise recommend using Fourier kernels for information dense data and audio and dilated kernels for images.
> > > >
> > > > In summary, there is **no one-size-fits all strategy to kernel selection**, but **propose to use Fourier kernels as a promising first choice** due its all round strong performance across different data modalities. We will make sure to emphasise this point as a strategy for selecting kernels in our paper.

---

> > > > > ### Author Response · Authors · 2024-08-13
> > > > >
> > > > > # W3
> > > > > We thank the reviewer for agreeing with our response in regards to baseline comparisons and we will add additional models and benchmark results to our paper where appropriate and possible, especially in regards to CHELA.
> > > > > ## Convergence Plots
> > > > > The convergence plots correspond to our ablation studies in Table 2, where we incrementally add all of design features that improve performance. Our results show that the incremental addition or our multi-resolution kernel design, low-rank Fourier kernels and BatchNorm for strucutral reparameterization dramatically **improve the speed of training convergence and performance** on both CIFAR and ListOps tasks.
> > > > >
> > > > > ## 'Why does BatchNorm (MRConv) achieve inferior training accuracy while yielding better testing accuracy in Figure 1(a)?'
> > > > > This is the result of the other models **overfitting to the training data**. Overfitting occurs when the model fits near perfectly to the training data and fails to generalise to the test data. Indeed both, dense and dense+multi-resolution convolutions over-fit and therefore whilst their training accuracy is high their test accuracy is low as can be seen in our convergence plots in our one-page attached pdf. On the other hand **BatchNorm MRConv fits the training data without overfitting**, resulting in **inferior training accuracy but superior test accuracy**. Further, MRConv without BatchNorm is shown to underfit, resulting in lower train and test accuracies as can be seen in the convergence plots. Fighting overfitting and improving generalization with our low rank kernels and multi-resolution structure is a key feature of our paper, and we will further add discussion to this point when adding the plots to our paper.
> > > > >
> > > > >
> > > > > We want to thank the reviewer for their valuable feedback, which we believe has significantly improved the paper.
> > > > > We kindly inquire whether all of their concerns have been resolved and if our replies and additional experiments have led them to change the paper’s score. If there are any remaining questions, we are ready to address them at the last minute. We look forward to their continued feedback to our response.
> > > > > Best regards!

---

> > > > > > ### Comment · Reviewer_8iUW · 2024-08-13
> > > > > > **Feedback to the Additional Response**
> > > > > >
> > > > > > Dear authors,
> > > > > >
> > > > > > Thanks for the further response with comprehensive details and extended experiments. Considering the authors' rebuttal and other reviews at the current stage, I believe this work has met the standard of NeurIPS, and I decided to raise my score. I suggest the authors to further polish the manuscript with these valuable explanations and additional results during the rebuttal period.

---

### Author Rebuttal · Authors · 2024-08-06

We thank all reviewers for their insightful and detailed reviews. The feedback provided has significantly helped to improve our work and solidified some of the claims in our paper. Please see the **attached one-page pdf for convergence plots and runtime experiments** as discussed in our rebuttals.

## Runtime Comparison
One consistent request among the reviewers was a throughput comparison of MRConv against existing efficient models with increasing sequence length. The efficiency of MRConv, both during training and especially at inference, is a key feature of our work. As a result, we have taken this request to heart and provided reviewers with an additional throughput plot in our one-page attachment. Utilizing optimized CUDA kernels provided by FlashFFTConv, our results show that MRConv is **substantially faster** than the efficient FlashAttention implementation, particularly for long sequences. Furthermore, our results emphasize that **even our non-reparameterized kernels are efficient and remain computationally faster than FlashAttention**, aligning with our theoretical complexity calculations. We thank all the reviewers who suggested such an analysis, as we feel it significantly improves our paper and demonstrates the performance of MRConv on long sequence modelling.

We hope that our adjustments and explanations adequately address the comments raised by the reviewers, resolving any concerns influencing their appraisal of our work and we invite any more feedback or points raised in the reviewer discussions.

---

### Author Response · Authors · 2024-08-12
**Final Enquiry on Reviewer-Author Discussion**

Dear reviewers,
we are entering the last day 2 days of the author-reviewer discussion period. We kindly ask the reviewers if our replies and additional experiments have addressed their concerns and if they might further change their score on the paper. If there are any additional questions, we will do our best to resolve them as soon as possible before the discussion period ends.
Thank you.

---

### Author Response · Authors · 2024-08-14
**Final Response to Reviewers**

We thank all the reviewers for their efforts in reviewing our paper and in contributing to the author-reviewer discussion. Your questions and suggestions have been insightful and thorough, and we sincerely believe that they have helped improve our work enormously. We look forward to incorporating your feedback into our paper, ensuring that it reaches its full potential. Your dedication and expertise are greatly appreciated, and we are grateful for the time and care you invested in this process.

Thank you once again for your valuable contributions!

---

### Decision · Program_Chairs · 2024-09-25

**Decision:**

Accept (poster)

**Comment:**

This paper introduces reparameterized multi-resolution convolutions (MRConv), a novel approach for parameterizing global convolutional kernels to enhance long-sequence modeling. The core idea is to decompose long convolutional kernels into a combination of kernels at multiple resolutions, each utilizing the same number of parameters. Experiments on the Long Range Arena, as well as on speech and image classification tasks, demonstrate the efficiency and effectiveness of MRConv.

All reviewers unanimously recognized the novelty and effectiveness of this work, resulting in its acceptance. The authors should revise their paper per the reviewers' suggestions for the final version.